# Discovery of SNP Molecular Markers and Candidate Genes Associated with Sacbrood Virus Resistance in *Apis cerana cerana* Larvae by Whole-Genome Resequencing

**DOI:** 10.3390/ijms24076238

**Published:** 2023-03-25

**Authors:** Aqai Kalan Hassanyar, Hongyi Nie, Zhiguo Li, Yan Lin, Jingnan Huang, Samuel Tareke Woldegiorgis, Mubasher Hussain, Wangjiang Feng, Zhaonan Zhang, Kejun Yu, Songkun Su

**Affiliations:** 1College of Life Sciences, Fujian Agriculture and Forestry University, Fuzhou 350002, China; 2191914004@fafu.edu.cn (A.K.H.);; 2College of Animal Sciences (College of Bee Science), Fujian Agriculture and Forestry University, Fuzhou 350002, China; 3College of Plant Protection, Fujian Agriculture and Forestry University, Fuzhou 350002, China

**Keywords:** sacbrood virus, single nucleotide polymorphism, larva, *Apis cerana cerana*, population differentiation index, nucleotide polymorphisms, variants, sequencing

## Abstract

Sacbrood virus (SBV) is a significant problem that impedes brood development in both eastern and western honeybees. Whole-genome sequencing has become an important tool in researching population genetic variations. Numerous studies have been conducted using multiple techniques to suppress SBV infection in honeybees, but the genetic markers and molecular mechanisms underlying SBV resistance have not been identified. To explore single nucleotide polymorphisms (SNPs), insertions, deletions (Indels), and genes at the DNA level related to SBV resistance, we conducted whole-genome resequencing on 90 *Apis cerana cerana* larvae raised in vitro and challenged with SBV. After filtering, a total of 337.47 gigabytes of clean data and 31,000,613 high-quality SNP loci were detected in three populations. We used ten databases to annotate 9359 predicted genes. By combining population differentiation index (F_ST_) and nucleotide polymorphisms (π), we examined genome variants between resistant (R) and susceptible (S) larvae, focusing on site integrity (INT < 0.5) and minor allele frequency (MAF < 0.05). A selective sweep analysis with the top 1% and top 5% was used to identify significant regions. Two SNPs on the 15th chromosome with GenBank KZ288474.1_322717 (Guanine > Cytosine) and KZ288479.1_95621 (Cytosine > Thiamine) were found to be significantly associated with SBV resistance based on their associated allele frequencies after SNP validation. Each SNP was authenticated in 926 and 1022 samples, respectively. The enrichment and functional annotation pathways from significantly predicted genes to SBV resistance revealed immune response processes, signal transduction mechanisms, endocytosis, peroxisomes, phagosomes, and regulation of autophagy, which may be significant in SBV resistance. This study presents novel and useful SNP molecular markers that can be utilized as assisted molecular markers to select honeybees resistant to SBV for breeding and that can be used as a biocontrol technique to protect honeybees from SBV.

## 1. Introduction

Both *Apis cerana cerana (A. c. cerana)* and *Apis mellifera (A. mellifera)* are widely used model species for apiculture and crop pollination, offering significant economic benefits [1,2]. However, concerns about honeybee health and its impact on the global economy have increased in recent years [2,3]. Sacbrood virus (SBV) was first reported as the most prevalent viral infection in honeybees in 1913, with White from the United States initially describing this group of diseases in *A. mellifera* [4,5]. SBV can infect larva, prepupa, and adult bees, but larvae are particularly sensitive to infection [6,7,8]. Due to its high replication rate, SBV can cause significant morphological changes in infected larvae, leading to their death and it becoming a widely distributed virus [9,10]. The disease was initially referred to as sacbrood due to the inability of infected larvae to pupate and the development of sac fluid, which is rich in SBV particles, beneath their undigested sack. SBV has been found to infect not only honeybees but also non-bee insects, such as bumblebees [11,12]. The diseased larvae exhibit a scale-like dark brown color resembling a ship, particularly in the mouth and digestive systems, which contain high amounts of external viruses [13]. Latent infection is significant for SBV propagation because it accumulates in the head and hypopharyngeal glands of infected nursing bees and is responsible for larval feeding through infected glandular secretions [14]. Honeybee diseases pose a significant threat to the health and well-being of honeybees, which researchers and beekeepers are seriously concerned about [15]. Honeybees, like other insects, have evolved innate immunity to protect themselves from external invaders. The immune system of *A. cerana* comprises 144 genes and 12 pathways [16]. SBV is a significant brood disease with seasonal prevalence and is much more lethal in *A. cerana* than in *A. mellifera* [17].

To develop effective disease control strategies for preventing SBV development, it is essential to gain a deeper understanding of bee resistance and the molecular mechanisms involved in resistant bee larvae. Various methods have been explored for managing SBV infection in honeybees, such as RNA interference (RNAi). However, large-scale RNAi isolation and purification are necessary to prevent SBV infection in bees [18]. The use of silver ions to protect bees from SBV is still challenging and not cost-effective [19]. Administering bee syrup mixed with herbal medicine may not be very effective in preventing SBV infection [20]. Requeening and feeding sugar syrup and pollen to the infected colony may improve the ability of the colony to manage the virus [21]. The application of immunoglobulin Y (IgY) to enhance immunization with a deactivated SBV vaccine for controlling Chinese sacbrood virus (CSBV) has proven useful but it requires large-scale purification [22]. Polypore mushroom mycelia have been shown to reduce viruses in honeybees, suggesting that fungi and their antibacterial compounds may add to bee SBV resistance. Silencing-related genes were examined in *A. cerana* using exogenous double-stranded RNA (dsRNA) produced by the pathogenic insect bacterium *Bacillus thuringiensis* (Bt), demonstrating the potential for using Bt as a platform for effective dsRNA production to control viral infections in host insects. Additionally, Radix isatidis extract has been shown to have antiviral properties against SBV [23].

Molecular markers and genomic DNA sequencing technologies are commonly used in studying honeybees from various perspectives. Genome-wide association studies (GWAS) in *A. cerana* have reported the usefulness of SNPs and simple sequence repeats (SSRs) as markers [24]. The *A. c. cerana* genome is smaller than the *A. mellifera* genome due to length inconsistency and has been sequenced to investigate adaptation and diversity in different climates [25]. Studies on disease resistance using SNP molecular markers in *A. mellifera* are more widespread than those in *A. cerana* [26,27]. The genomes of both species have been sequenced and completed [28]. The first draft genome of *A. cerana* was generated in 2015 by Park et al., with an estimated size of 228.32 Mb, 2430 scaffolds, and a 152.0 N50 genome coverage [29]. In 2018, Diao et al. reported the genome assembly of the Chinese honeybee species *A. c. cerana*, with a genome size of 228.79 Mb [16]. *A. c. cerana* is suitable for studying population differentiation index (F_ST_), gene variations, and genetic structure. Illumina sequencing is a precise platform that can generate millions of short-read sequences [30]. DNA resequencing using SNPs with F_ST_ and population differentiation index (π) has been used to assess the high royal jelly production of the honeybee *A. mellifera* [31]. SNP molecular markers have also been used to examine genetic variations in honeybees that have undergone Africanization for ancestry assignments. F_ST_ comparison pools and individual sequencing have been reported for studying the populations, structure, and diversity of European honeybees [32]. SNP markers are a contemporary approach to molecular breeding programs, especially in disease control and genomic research. More studies on *A. c. cerana* are required to identify which SNPs alter cellular function and contribute to disease to cover any protection gaps for this bee species. SNP molecular markers are commonly used in genetic research in various organisms [33].

All studies have shown that identifying SNPs and molecular mechanisms of larval resistance to SBV is crucial for disease control in honeybees. However, the exact mechanism by which SNPs and alleles affect honeybee susceptibility or resistance to *A. c. cerana* remains unknown. To address this, we conducted experiments to explore the molecular mechanism of larval bee resistance to SBV. We raised larvae in vitro and challenged them with SBV, and the results are shown in Figure 1. Genomic DNA was extracted from 90 individual R and S larvae using next-generation high-throughput DNA sequencing, and an integrated analysis was conducted based on a comparison of R and S larvae to screen SNP molecular markers and their target genes related to SBV resistance. The analysis included resequencing of 90 samples, data evaluation, genome comparison, mutation detection and annotation, and genetic evolutionary analysis based on variant detection results. We analyzed differences in SNPs in honeybee larvae and their target genes and investigated the distribution of allele frequencies between R and S in the larvae. We found several regions with SNPs and their target genes that were significantly associated with SBV resistance, and a summary of the results is shown in Table 1. A minor allele frequency (MAF < 0.05) analysis was conducted between R and S to assess variations compared to the reference genome, and a high percentage of genotypes was selected to verify the candidate SNPs. Among them are two SNP loci located on the 15th chromosome that were significantly associated with SBV resistance based on their allele frequencies. We validated each SNP with multiple confirmations including (at the same colony level, field validations, comparisons of colony resistance (CR) and colony susceptibility (CS), long-distance geographical locations, and tested on resistant queens).

The primary aim of this study was to identify SNPs linked to SBV resistance in honeybees and to explore genomic DNA variations between R and S larvae using whole-genome resequencing. Additionally, this study aimed to identify SNP loci, genotypes, and associated genes to gain a better understanding of the molecular mechanisms underlying SBV resistance in honeybee (*A. c. cerana*) larvae. The results of this study may be useful for accelerating molecular breeding, assisted selective breeding to develop SBV resistance, and as a biocontrol method to manage SBV infection in honeybees.

## 2. Results

### 2.1. Confirmation of the SBV Purity and Larval Mortalities

The purity of SBV was validated through multiple methods, including reverse transcription-polymerase chain reaction (RT–PCR), SBV detection, transmission electron microscopy (TEM) characterization, sequencing, and phylogenetic analysis. The results revealed that SBV was the only infection detected in the pure viral samples, and none of the six common bee viruses (acute bee paralysis virus (ABPV), Kashmir bee virus (KBV), black queen cell virus (BQCV), Israeli acute paralysis virus (IAPV), and chronic bee paralysis virus (CBPV)) were detected. Colony screening for common bee viruses showed single virus infection, multiple virus infections, and no infection (Figure 1A). The q-PCR analysis of viral particles in the purified samples yielded 6.32 × 10^12^ copies/μL, and the generated standard curve detected SBV. The phylogenetic analyses are shown in Appendix A–D. Inoculated larvae clearly exhibited SBV symptoms, while pupae were uninfected (Figure 1B). To assess the efficient infectiousness of the purified virus, a comparison of the survival time between the treated (inoculated with SBV) and control groups (virus-free) in three colony larvae were conducted. The results showed a significant difference (Figure 1C); however, the comparisons of the control within the group were not significantly different (*p* > 0.01). The comparison of survival function between the treatment and control groups of resistant (CR) and susceptible (CS) colonies showed a difference. However, the mortality rates of the control groups between CR and CS were not significantly different (Figure 1D). 

### 2.2. Resequencing and Data Quality Statistics Compared with the Reference Genome

The resequencing of 90 individual samples from *A. c. cerana* S and R larvae was completed, and the results are presented in Appendix A. The amount of data analyzed was 337.47 gigabytes, with an average Q 30 score of 92.73%. The average mapping ratio of the samples to the reference genome was 87.92%, with an average coverage depth of 14 x and genome coverage of 99.04%. The statistics for data evaluation, sequencing, data volume, sequencing data quality, GC content, genomic comparison, comparison rate, genomic coverage, and genome coverage depth are displayed in Appendix A. The whole-genome resequenced clean reads of R and S samples were compared using the same species, and the average mapping ratio was relatively high, with an average of 90 samples at 87.92%, Appendix A. Comparisons between R and S and to the reference genome of *A. c. cerana*, all variants of SNP loci, and Indels are summarized.

### 2.3. Identified SNP and Indels

The pipeline was utilized to identify all variations through data processing, bioinformatics analysis, and functional annotation, as shown in Appendix A. After sequence alignment and a comprehensive analysis, a total of 31,000,613 SNPs and Indels (Appendix A) were detected in the three populations, consisting of 15,596,070 SNPs in R and 15,404,543 SNPs in S. The DNA mutations were categorized into transition and transversion (Ti/Tv). In the R group, the Ti and Tv of SNP were 12,872,420 and 2,723,650, respectively, composed of 9,292,227 heterozygosity and 6,303,843 homozygosity, with an average of 59.56% heterozygosity. In the S group, the Ti and Tv of SNP were 12,713,876 and 2,690,667, respectively, including 8,871,581 heterozygosity and 6,532,962 homozygosity, with an average of 56.88% heterozygosity (Appendix C, Table A1 and Table A2). These results indicated differences in SNP numbers and mutations detected in the two groups. To ensure the reliability of the sample SNPs, the number of reads supporting the detected SNPs and the cumulative statistical distribution of the distance of adjacent SNPs are shown in Appendix A. Genome-wide SNP mutations showed that two types, C: G > T: and T: A > C: G, had the highest SNP mutations, while two other types, T: A > G:C and C: G > A: T, had the lowest. The other two mutation types, T: A > G: C and C: G > A: T, had almost the same number of SNP mutations between the 90 samples, as shown in Appendix A. Based on the comparison results of the samples with the reference genome, all the different variants between the samples were summarized, and the DEG SNP list file format is shown in Appendix A. The unique SNP genotypes after filtering are shown in Appendix A. Private SNPs in the R group contained 275,421 SNPs (Appendix A), and 286,551 SNPs were detected in the S group (Appendix A).

### 2.4. SNP and Indels Annotation

Based on the location of the mutation site in the reference genome and the location of the gene in the reference genome, we can identify the region of the genome, including intergenic regions, coding sequence regions (CDS), and the impact of the mutation, such as synonymous and nonsynonymous mutations. Using the VCF format file as both input and output, we can assess various fields in the info column of the VCF file, including EFF = Effect (effect impact, functional class, codon change, amino acid change, amino acid, length gene name, transcript, biotype gene coding, transcript ID, exon rank, genotype number, and error warning), which are added as part of the output. All 1,048,575 SNPs and 424,131 indels identified were annotated, and gene information was obtained from the reference genome to detect and annotate the location of small indels in CDS regions. The results show that most of the SNPs and indels were annotated in intergenic regions. Figure 2 displays the annotation statistics for all SNPs and indels, and Appendix A provide detailed annotations of the SNPs and indels across CDS and the genome for the 90 samples.

### 2.5. Variation in the Gene at DNA Level Analysis

Variations in the CDS may lead to changes in gene function and genes with functional differences between the sample and the reference genome. We can find genes that may have functional differences by looking for CDS and variant region (SV) nonsynonymous mutated genes between the reference genome and the sample. The differences in genes between the R and S individuals of 90 samples compared with the reference genome are shown in Appendix C, Table A3 and Table A4. Significant regions and predicted regions were detected based on the combination of F_ST_ and compared with the reference genome, and INT < 0.5 and MAF < 0.05 with their significant region genes are shown (Appendix A).

### 2.6. Annotation of Variant Genes

The Basic Local Alignment Search Tool (BLAST) was used to compare mutant genes with functional databases [34] such as SwissProt [35], the Gene Ontology (GO) database [36], Clusters of Orthologous Groups of proteins (COG) [37], and the Kyoto Encyclopedia of Genes and Genomes (KEGG) [38]. The annotations of the genes were obtained from these public databases to analyze their functions. The annotation results of all genes with a comparison of the mentioned databases and the integrated functional analysis for all genes are shown in Appendix A. Functional and enrichment analysis of significant SNPs and their significant target genes based on the combination of F_ST_ and π from COG, KOG, GO, and KEGG and the results of the classification are shown in Appendix A and Appendix A. 

### 2.7. Genetic Evolution and Diversity

To investigate the evolutionary relationships between R and S from the same species, a phylogenetic tree was constructed based on a neighbor-joining (NJ) tree composed of 45 R and 45 S samples from three populations of all informative SNPs used in the phylogenetic analysis. The results showed that the group samples from three colonies were well differentiated in clades; however, the R and S from the sample colony were still slightly mixed together and differentiated from other colonies. Therefore, the three populations were the same species and in the same location, and there was no distance between the sampled geographic areas. However, the phylogenetic tree of SNP validation in the field from different geographic locations and a comparison of RC and SC showed a distinct evolution between R and S larvae, which were well differentiated into separate clades and branches (Appendix B, Figure A1 and Figure A2).

### 2.8. Population Structure

A population structure analysis can quantify the number of ancestors of the studied populations and infer each sample’s blood origin. It is currently a more widely used group cluster analysis method that helps us understand material evolutionary processes. The number of subgroups (K-value) was set from 1 to 10 for the clustering of the study population. The optimal number of groups was determined based on the valley value of the cross-validation error rate in the admixture, and the individual ancestries from SNP genotypes were investigated. The clustering results of K-values from 1 to 10, the cross-validation error rate corresponding to each K value, and the model K = 6 obtained from 90 samples in the two groups is shown in (Figure 3C,D).

### 2.9. Principal Components Analysis

Through a principal component analysis (PCA), it is possible to determine which samples are relatively close and which samples are relatively distant, which can assist in understanding the evolution between traits. The results showed that total of 20.76% variance between R and S. PCA 1, PCA 2, and PCA 3 showed some differences. Most of the SNPs in R were relatively separately clustered; however, simultaneously, the clusters were close to S in distance. A similar result was observed in the phylogenetic tree PCA 3D plot (Figure 3E). To estimate the relative kinship between two individuals in a natural population we use the affinity metric. The affinity itself is a relative value that defines the genetic similarity between two specific materials and the genetic similarity between any material, so when the kinship value between the two materials is less than 0, it is directly defined as 0. The frequency distribution of the kinship value is shown in Appendix A. 

### 2.10. Selective Sweep and Selection Signature

In the selective sweep analysis, due to the selection-specific site (natural selection/manual selection), the frequency of the neutral linkage relationship of neutral mutation increased. In contrast, the frequency of no linkage neutral site mutations decreased, resulting in a decrease in polymorphism in this region. The method for detecting selective sweep regions in the genome-wide range is to calculate the population genetic indicators of all SNP sites within a sliding window (e.g., 100 kb). A specific step size (e.g., 10 kb) was chosen, such as F_ST_ and θ_π_ based on the calculation results, and a particular region was selected (Appendix A). The samples were divided into two groups for selective sweep analysis, and the combination of F_ST_ and θ_π_ from the selective sweep analysis in the two groups is shown in Figure 3F. Table 2 and Table 3 show the distributions of G versus C and C versus T allele frequencies of SNP KZ288474.1_322717 and KZ288479.1_95621, which were selected and validated, and colonies were selected to validate these two SNPs (Table 4).

### 2.11. Significant Regions between Resistance and Susceptibility

Based on the comparison results between R and S using multiple tests of F_ST_ and θ_π_, independent and combined F_ST_, and θ_π_ genome-wide analysis of each candidate site showed that R and S are considerably different from the reference genome in the 457 scaffolds region of the chromosome-level selective sweep analysis. Comparison results of each chromosomal region were extracted at 19,551 positions. A selective sweep analysis showed that the window and step size statistics intersection in 6,464 scaffolds between R and S were significantly different (Figure 4). Functional and enrichment analysis results of the selected regions are shown in Figure 5.

Box plots of F_ST_ and π R and S selected genomic regions with selective sweep signals versus the whole genome results are shown in Figure 4. The box represents the range of quartiles (IQR) between the first and third quartiles (25th and 75th percentiles, respectively), and the line in between represents the median. Whiskers represent the lowest and highest values within the first and third quartiles (1.5 × IQR). The outliers beyond whiskers are shown as hollow dots. A Mann–Whitney U test was used to calculate statistical significance, with a *p*-value of <0.05 considered significantly different.

### 2.12. Flanking SNP between Resistance and Susceptibility

Due to the large number of SNPs, we considered narrow significant regions, and a high percentage of allele frequency was selected. Additional statistical analyses were conducted for the best difference between R and S compared with the reference genome. F_ST_ and π S divided the top 1% R 84 SNP positions, and F_ST_ π R divided the top 1% S 735 SNP positions (Appendix A). When F_ST_ π S was divided by R and F_ST_ π R was divided by the S top 1% or >0.99, the best difference of 10 and 19 SNPs, respectively, was significantly different.

### 2.13. Verified SNPs

To verify the candidate SNPs, we selected 10 SNPs based on the combined F_ST_ and π S divided R top 1%, and 10 SNPs based on the combined F_ST_ and π R divided S top 1% with a high percentage genotype. After verifying the first candidate SNPs, none were found to be significantly different. We then performed SNP validation for the second group of candidate SNPs. Among them, two SNPs (GenBank IDs and positions: KZ288474.1_322717, G > C, and KZ288479.1_95621, C > T) and their corresponding alleles (based on their allele frequencies) were significantly different in the three colonies with the same colony samples used for DNA sequencing. Further SNP validation was performed to ensure the reliability of these two SNPs with comparisons of R and S larvae from different geographic locations and populations. The results showed a highly significant difference between the R and S larvae. 

According to SNP annotation, the first SNP (KZ288474.1_322717) was an upstream effect transcript ID = RNA 9701 with a gene close to the *APCC*_06899 gene at 1804 bp distance. The second SNP was an intergenic SNP relatively far from the *APCC*_01833 gene with a 53,591 bp distance. 

The comparison results of multiple SNP validations from the same individual sample and different samples from different geographic locations were consistent between two SNPs in the R, S, and genotyped queens conformed to each other with a slight change.

In addition, in the R samples, the frequencies of G allele at the SNP KZ288474.1_322717 position and C allele at the SNP KZ288479.1_95621 position were high percentage allele frequencies. On the other hand, in the S samples, high percentage allele frequencies of C and low percentage allele frequencies of T were observed; however, in the R samples, C and T at two SNPs position were very low and even were not observed in some colonies. A comparison of two SNPs with same sample was conducted, which indicated the reliability of each SNP and the accuracy of the Sanger sequencing method and supported the reliability of the first SNP detection and validation results.

## 3. Discussion

In this study, we challenged larvae of honeybees (*A. c. cerana*) with SBV from 90 individual samples from three populations in Fuzhou, China, then resequenced and analyzed them to screen SNP genetic markers associated with SBV resistance [28,29,32]. The samples were resequenced using an Illumina Nova His Seq X-Ten for sequencing (150 bp PE), and 31,000,613 SNPs were identified, including 18,163,808 (42%) homozygous SNPs, 12,836,805 (58%) heterozygous SNPs, 5,414,317 transversion SNPs, and 25,586,296 transition SNPs, and 9359 predicted genes from 90 individuals were annotated [39]. To identify genetic markers associated with SBV resistance, we compared the VCF list of 179,7078 SNPs between R and S, which was identified by GATK [40], with a comparison of the reference genome INT < 0.5 and MAF < 0.05 using multiple tests, especially F_ST_ and π, independently and in combination using Popgenome [40,41]. We determined significant regions of each SNP and their target genes at the level of the entire genome chromosome and the relevant sections at each chromosomal level. After identifying 6464 relevant significant regions at the scaffold level between R and S, we combined the F_ST_ and π ratios. All regions were significantly different for segregation sites and the INT was 0.05 and the MAF was < 0.05 for R and S. The significant regions under artificial selection with F_ST_ and π 0.99 and 0.95 (top 1% and top 5%) were then selected by integrating the F_ST_ and π statistics. We cataloged 84 unique SNPs with seven predicted genes and 735 unique SNPs, with ten anticipated genes based on combined F_ST_ and π. F_ST_ and π R divided S and F_ST_ and π S divided R were significantly detected in 90 samples.

To identify and validate the SNPs of interest, we selected the top ten SNPs with the highest proportion of genotypes based on the F_ST_ and π analyses, specifically the F_ST_ and π R divided S top 1% and the F_ST_ and π S divided R top 1%. Among these, we found two significant SNPs, KZ288474.1_322717 (G > C) and KZ288479.1_95621 (C > T), located on the 15th chromosome with GenBank IDs, which were associated with SBV resistance based on their allele frequencies.

The validation of the SNPs began with 16 R and 16 S samples from three colonies, followed by validation with 32 R and 32 S samples from the same colonies using DNA sequencing. Out of the 20 candidate SNPs, three were not well amplified, and 15 were not significantly different, shown in Appendix A. For field validation, one SNP (KZ288474.1_322717) was selected from 32 R and 32 S regions of three asymptomatic colonies in Fuzhou. During larval rearing, we predicted that the three colonies might be resistant to SBV, as only a few larvae were infected with the virus at the same dose. The experiment was repeated with a double dose of the virus, and after sequencing, evaluation, and analysis, no significant difference was observed between R and S in allele frequency (*p* > 0.05), indicating that the three colonies were resistant to SBV. For the second SNP (KZ288479.1_95621), three symptomatic colonies were selected in Minhou after larval rearing, sequencing, and analysis. The allele frequency was significantly different. During the validation process, it was discovered that the honeybee colony might have evolved naturally to become resistant to SBV and not carry the susceptible allele C/C at the SNP KZ288474.1_322717 position. Each colony or population exhibited different levels of resistance and susceptibility [27].

Therefore, to further validate the SNPs and assess their association with SBV resistance across different geographic locations, we selected additional colonies from various regions. The SNP validation data showed significant variations in allele frequency between 32 R and 32 S samples in each symptomatic colony. Asymptomatic colonies had high frequencies of G/C and C/C alleles but very low frequencies of G/G alleles, while symptomatic colonies had low allele frequencies. In addition, asymptomatic colonies had a high C/C allele frequency at both SNPs, whereas symptomatic colonies had a high T/T allele frequency at the second SNP (C/T). To investigate whether these genotypic and phenotypic characteristics were consistent across different geographic regions, we obtained symptomatic samples from three colonies in Yunnan Province, located 2306.5 km away from Fuzhou City. Although these colonies were infected with DWV and did not show visible SBV symptoms, we selected R samples (pupae) for analysis. Screening and analysis of both SNPs revealed that most of the pupae samples had G/G and C/C alleles, and susceptible alleles (C/C and T/T) were not detected. These results provide further evidence of the association between these SNPs and SBV resistance, regardless of geographic location. 

Additional evidence was gathered from 14 resistant queens from Nanping and three apiaries, as well as one apiary in Fuzhou. Genotyping revealed that only one queen carried the G/C and C/T genotypes for the two SNPs, while the remaining queens had G/G and C/C genotypes. These results, combined with those of the genotyped queen, suggest that honeybee colonies with homozygous resistant G/G alleles at the SNP position KZ288474.1_322717 and C/C alleles at the SNP position KZ288479.1_95621 are likely to be strongly resistant to SBV. Conversely, colonies with homozygous susceptible alleles (C/C and T/T) are expected to be extremely susceptible to SBV, while those with heterozygous susceptible alleles (G/C and C/T) may only be moderately susceptible. These findings underscore the importance of choosing colonies and queens for breeding programs.

The fact that the alleles generating SBV resistance in both SNPs were found to be correlated with one another in the 90 samples of genotype sequences and SNP validation highlights the accuracy of these SNPs. To further ensure their accuracy, we compared the results of 210 samples for the two SNPs using the same sample, as well as 48 samples from two different sequencing companies. We aimed to verify whether the G/G allele at SNP position KZ288474.1_322717 and C/C allele at SNP position KZ288479.1_95621 corresponded with one another, and vice versa. We found that the results from both groups were nearly identical after peak assessment, indicating that sequencing errors were minimized. We further validated the successful SNPs in a large number of samples from various geographic locations to confirm their reliability. The use of these SNPs as valuable genetic markers in selective breeding programs for improving multiple honeybee traits, including SBV resistance, has been suggested in previous studies that found the same corresponding alleles related to SBV resistance, high royal jelly production, and chalk-brood resistance in *A. melliferra* [26,27,31].

Late SBV infections were observed in the prepupae during the larval-rearing experiment. The larvae and prepupae were not identified as SBV-resistant until the larvae had fully transformed into pupae at the 12–14 day instar and were then individually sampled as R [42]. The mortality rate of larvae in the susceptible colony was high (39%), in contrast to the low mortality rate in the resistant colony (28.12%) after inoculation. The survival rates of the control groups in the CS and RC were relatively high, at 85% and 88%, respectively [43]. 

### Population Genetics and Variations

The clean reads were mapped to the reference genome to obtain the mapped reads. A high sequencing read ratio (~70%) indicates appropriate reference genome selection and the absence of contamination during the experiment. The assembly of the reference genome is important for the quality of sequencing reads and the locatability of reads in the reference genome, which is directly proportional to the degree of similarity between the species. On average, the mapping rate of the sequence data was 87.92% [44]. Compared to SNP variation, indel variation was generally less frequent. Most SNPs/Indels were annotated as intergenic variations, with fewer SNPs found in introns. A detailed annotation of SNP/Indels is provided in [45]. 

Whole-genome sequencing data were utilized to investigate the evolutionary relationships among three populations using various analyses, including phylogenetic tree, population structure and admixture, PCA, and kinship analyses. These analyses were conducted to describe the taxonomic and evolutionary relationships between the traits that were assumed to have a common ancestor. The three populations showed clear differentiation in clades with some mixing within groups. Additionally, some distance was observed in the ancestries between the three populations, and the best K = 6 was obtained in admixture analysis [39]. 

The mRNA level genes were evaluated using Illumina sequencing data. Selective sweep, INT < 0.5, and MAF < 0.05 were applied to obtain highly consistent genes. All genes within 100 kb of significant SNP regions were extracted, and their annotations were considered. All predicted gene directions were upregulated. The top 1% of genes based on combined F_ST_ and π were selected for SNP validation, considering the narrow range of significant regions and the high allele frequency. The SNP genotypes were present in the R group, and they differed from the selected verified reference genome alleles in the S group. However, only a few genes were annotated, which could not explain the more positive effects; therefore, we considered the top 5% annotation results from the same site where we detected two SNPs [31].

SNP and indel annotation, as well as a prediction of different genes, were performed on individual samples from R and S, comprising 90 annotated samples, and on the specific region detected by the combined F_ST_ and π top 5%. The functional annotation of 90 individual samples was conducted across 10 databases of 9,359 different genes, including integrated functional annotation statistics, GO gene list 1399, function, 1812 in the biological process (BP), 820 in cellular component (CC), and 10,583 unique genes in molecular function (MF). In the 116 KEGG pathways, 2908 and 3842 unique genes with positive effects were annotated, most of which were involved in hypothetical protein signal transduction mechanisms [45,46]. Some of the most enriched pathways in KEGG were spliceosome (111 genes), RNA transport (105 genes), purine metabolism (103 genes), ribosome (98 genes), endocytosis (97 genes), carbon metabolism and protein processing in endoplasmic reticulum (96 genes), ubiquitin-mediated proteolysis (82 genes), and phagosome (55 genes). The other genes involved in different pathways totaled than 50 genes. Some of these pathways have a significant effect on disease response [47,48,49,50]. Based on the functional annotation of 90 individual samples compared to the reference genome of *A. c. cerana*, 9359 different predicted genes against ten databases were annotated, which is close to the previously identified and annotated genes in the *A. cerana* genome (9627) [51] 

Based on the F_ST_ and π values, R and S were divided into the top 5% gene annotations, resulting in 119 genes, of which 116 were annotated in various databases, such as COG, GO, Nr, Nt, Pfam, SwissProt, and TrEMBL (Appendix A). Out of the 844 genes annotated for biological processes, 49 were related to single-organism cellular processes (GO:0044763), 28 were associated with transport (GO:0006810), and 51 were involved in metabolic processes (GO:0008152), with the first two categories being statistically significant (*p* < 0.05). Notably, *APICC*_03984 was found to be involved in the activation of MAPKK activity, pathogenesis, cellular hyperosmotic response, and signal transduction involved in filamentous growth in the biological process category. The curated GO BP annotation dataset shows that 63 genes participate in the activation of MAPKK biological processes, such as HRAS, JAK2, and BRAF. In the cellular component category, 79 out of 520 genes were related to the cytoplasm (GO:0005737), with the most enriched genes involved in metabolic processes (10), catalytic activity (10), cellular process (7), binding (10), biological regulation (5), response to stimulus (3), and signaling (2). The other genes were involved in different positive effects, and no genes were involved in the death pathway. *APICC*_00122 was found to be involved in exopeptidase activity. While the insect midgut has been primarily associated with digestion and detoxification, endopeptidases such as serine proteases (trypsin and chymotrypsin-like) and exopeptidases with varying terminal amino acid specificity (aminopeptidases and carboxypeptidases) are believed to play a key role in protein hydrolysis [52]. 

In the GO molecular function category, two genes were significantly enriched in ATPase activity (GO:0016887). Among the 90 genes in KOG, 46 were annotated in KEGG, with ABC transporters (*APICC*_00398 and *APICC*_04707) having the highest enrichment factor (14.77) and a q-value of 0.052. Other enriched pathways included beta-alanine metabolism (*APICC*_01290 and *APICC*_10032), lysine degradation, glycerophospholipid metabolism, phagosome (*APICC*_03257 and *APICC*_08196), fatty acid metabolism (*APICC*_10032), protein processing in the endoplasmic reticulum, and endocytosis (*APICC*_01294 and *APICC*_10111) [53,54].

The genome of *A. cerana* has been previously reported to have a de novo assembly with 10,182 predicted genes [29]. Another study reported the genome sequencing and assembly of *A. cerana* with a data size of 228 Mbp and 879 scaffolds [17]. The *A. c. cerana* genome still needs further sequencing and improvement. Although the genomic size of *A. cerana* is slightly smaller than that of *A. mellifera*, more evidence is required to explain the significant differences between the two species [24]. 

Bee viruses are usually asymptomatic but can significantly affect honeybee health and their lifespan under certain conditions. While SBV symptoms in larvae are evident, they are not clear in adult bees [55,56]. Worker bees with hygienic behavior can help prevent pathogen dispersal by removing infected larvae, but this behavior can cause infections in other bees, leading to virus transmission through feeding into the larva [57]. Although technical bee management methods, such as requeening, replacing the old brood comb, and using herbal medicines, can help control SBV, they are not very efficient. It is necessary to manage bee colonies to reduce the levels of Nosema disease and control Varroa mites throughout the year [19]. The SBV isolated in this study is valuable for future research on honeybees. Researchers can use pure viruses to expand the study of SBV resistance, study gene function and characterization, and apply CRISPR–Cas9 to edit genes closely related to SBV resistance. Other potential projects include transcriptome analysis of larvae using purified virus, genotyping using identified SNPs to inoculate the larvae, RNA sequencing, applying SBV to different bee species, and studying the behavior, physiology, and other aspects of honeybees.

## 4. Materials and Methods

### 4.1. Colony Selection

Experimental colonies were established at the experimental apiary at the College of Animal Sciences (College of Bee Science), Fujian Agriculture and Forestry University, to conduct experiments. Some samples were directly collected from apiaries. The samples and colonies were selected from Fujian Province and Yunnan Province. The details of sample collection and the number of colonies used in this study are summarized in Table 4.

**Table 4 ijms-24-06238-t004:** Experimental colony details used in this study.

Date	Apiary Location	Purpose	No. of Colonies	SNP 322717	No. of Colonies	SNP 95621
2017	Yongtai, Fuzhou	Virus purification	4	250	4	1000
	Minhou	Healthy colony selection	8	15	8	120
	Minhou	Larval rearing and genomic DNA sequencing	3	90	3	90
2021	Fujian and Yunnan Provinces	SNP validation	32	926	34	1022

To purify SBV from diseased larvae and to select healthy colonies, this study was conducted in different geographic locations in Fujian Province. We selected three apiaries in Yongtai and one apiary in Fuzhou. A symptomatic colony was sampled from each apiary. Three frames with obvious SBV symptoms were selected from each colony and extracted from the hives. The frames were marked and immediately placed on dry ice. The samples were quickly transferred to the laboratory. From each colony, 250 old, infected larvae were collected in pooled samples. All the samples were directly stored at −80 °C for isolation and purification of SBV.

Three out of eight healthy colonies were selected for larval rearing after screening against seven common bee viruses, see the details in the virus isolation and purification section. After larval rearing from three colonies challenged with SBV 90 samples, each colony was composed of 30 samples (15 S larvae between 7–10th instars and 15 R larvae (pupae) at 12–14th instars) randomly selected for genomic DNA extraction and genome sequencing. The first SNP validation was conducted using the same colony samples that were previously used for genomic DNA sequencing, which were composed of 16 R and 16 S. The second SNP validation of each colony was composed of 32 R and 32 S from the same three colonies for SNP 95621 and SNP 322717; two colonies from the same colonies and one extra colony from Fuzhou were used.

Three asymptomatic bee colonies from the Fuzhou area were selected for in vitro culture to verify the effectiveness of these two SNPs in the field. Each colony consisted of 64 samples (32 R and 32 R). By SNP verification, three asymptomatic colonies in Fuzhou and three symptomatic colonies in Minhuo were compared. The SNP 95621 was validated in two symptomatic colonies in Putian, one asymptomatic colony in Fuzhou, and three symptomatic colonies in Minhou County. Each population consisted of 64 larval and pupal samples (32 R and 32 S) of SNP 322717.

To verify whether these two SNPs function reliably in different geographical locations, we conducted further SNP verification in three symptomatic colonies (Nanping, Putian, and Fuzhou) and selected SNP 322717. Another SNP (95621) was identified in two symptomatic colonies in Putian, one colony in Nanping, and two colonies in Fuzhou. Each colony consisted of 32 R and 32 S. Samples in frame form, one symptomatic and two asymptomatic from Yunnan Province, were obtained by express, and 48 R samples (pupae at the white eye stage) were randomly sampled from combs and analyzed.

Further SNP validation in resistant queens was conducted on 14 queens from four apiaries (three apiaries in Nanping: twelve queens and two apiaries in Fuzhou: two queens), which were directly collected from the hives, kept in queen cages, and transferred to the laboratory. In addition to comparing the genotypes, we investigated whether the allele frequencies of the two SNPs corresponded to each other and whether the Sanger sequencing was accurate. Of 210 samples from R and S, the queen samples included in SNP validation were compared separately. Overall, a total of 926 samples for SNP 322717 and 1022 samples for SNP 95621 were used for SNP verification in this study.

### 4.2. Isolation and Purification of Sacbrood Virus

To isolate SBV, all necessary materials were autoclaved before virus purification. Prior to purification, 15 infected larvae were randomly selected. Additionally, 15 uncapped larvae were sampled from eight candidate colonies. Total RNA was extracted, cDNA was synthesized, and seven common bee viruses were screened using the RT–PCR method, as described by Hassanyar et al. [58]. After screening, two colony samples that were positive for SBV and negative for the other viruses were used for SBV purification. We detected other viruses (DWV and IAPV) in the two samples, including SBV, and these samples were discarded. To purify SBV, we followed the protocol described by Maori et al. with some modifications [59]. Briefly, all the samples were separately ground into powder using liquid nitrogen with a mortar and pestle and then homogenized in 0.01 M Na-phosphate (pH 7.6) and 0.2% Na-deoxycholate (pH 7.2) (Sangon Biotech, Shanghai, China). The samples were transferred into centrifuge tubes (Axygen, USA) and centrifuged at 800× *g* for 20 min. The supernatants were collected in new clean tubes and the pellets were discarded. The high-speed centrifuge (HITACHI ultracentrifuge CPN 80X, Japan) was turned on for a while to cool to 4 °C. The supernatant was transferred into ultracentrifuge tubes and balanced before centrifugation at 27,000× *g* for 4 h. The supernatants were discarded, the pellets were dissolved using 0.4% sodium deoxycholate acid (Na-DCA), and 4% Brij 58 (Sangon Biotech, Shanghai, China) was added, vortexed, and centrifuged again at 800× *g* for 15 min for clarification. The supernatants were carefully transferred into the same tubes we used before, then 5.5 g cesium chloride (CsCl) was added to each tube, well-mixed, then balanced, and transferred into the ultracentrifuge again and centrifuged at 27,000× *g* for 24 h at 18 °C. Two typical cloudy bands were formed in the supernatant and in the middle of the tubes. The bands were carefully collected separately in fresh tubes. To remove CsCl from the solution, three dialysis cycles were performed in double-distilled water (ddH2O) at 4 °C, and the water was replaced with fresh water after eight hours. Then, the solution was prepared in different vials (250 mL, 500 mL, and 1000 mL) and kept at −80 °C for viral load quantification and inoculation.

### 4.3. Confirmation and Verification of SBV Purity

The first observation of SBV was conducted using TEM, following the protocol developed by Zhou et al. [60]. Briefly, 20 µL of the two purified samples was placed in separate fresh tubes, kept on ice, and transferred to the College of Plant Protection, Fujian Agriculture and Forestry University, for observation under TEM. The grids were negatively stained with 2% sodium phosphotungstate at pH 6.8 for 2 min. To prepare the standard staining solution, we used 1% phosphotungstic acid (PTA) at pH 7.2. A drop of the viral suspension was applied onto a formvar- or collodium-covered electron microscope grid for 30 s, and the liquid was absorbed using filter paper. A drop of the staining solution was used after a few seconds, then absorbed and dried. The specimens were placed into an electron beam and then transferred to the TEM room for characterization. The TEM (model HITACHI-H 7650 Japan) was used to observe the particle characteristics of SBV in the two samples. The particle size of SBV was measured and imaged at 100 nm. RT–PCR was used to conduct the second verification of SBV with specific primer pairs [61], and the third verification was conducted by Sanger dideoxy sequencing and phylogenetic analysis.

### 4.4. Quantitative PCR Assays of Purified Virus

qPCR was performed to measure the exact copy number of the virion particles [62]. Purified SBV was quantified on a high-speed cryogenic centrifuge (5840R; Sigma) using TransStart^®^Top Green qPCR SuperMix + Dye II (Beijing Quanjin, China). The cDNA was diluted 5 and 10 times, and templates of plasmid (Chinese sacbrood virus) CSBV DNA were used to create a standard curve and serially diluted from 10^−2^ to 10^−8^. cDNA and plasmid CSBV were used as templates for qPCR. The qPCR solutions were prepared according to the following components, and the total reaction system was 20 µL: cDNA purified virus 2 µL, Chinese Sacbrood virus (CSBV) forward primer 0.4 µL, reverse primer 0.4 µL, 2 × TransStart^®^ Top Green qPCR SuperMix 10 µL, and ddH2O 7.2 µL. qPCR amplification consisted of 40 cycles of 95 °C for 30 s, 95 °C for 5 s, 58 °C for 15 s, 72 °C for 10 s, and 95 °C for 0.5 s. Data were analyzed based on the molecular weight and the CSBV DNA plasmid sequence length [63].

### 4.5. Larval Rearing In Vitro

Larval rearing was conducted between Mar and Jun 2018 and continued until Nov 2018. This was based on the protocol developed by Crailsheim et al. [64]. Larvae were reared in a 48-well cell culture plate (clusters of sterile non-pyrogenic polystyrene; USA). To collect same-age larvae, an empty comb was placed in the hive for 24 h to allow the worker bees to clean up. The comb was inserted into the queen cage and the queen was confined in the cage for 24 h and placed in the middle of the hive (the queen bee and larvae could be better fed by worker bees). After 24 h, the eggs produced by the queen bees in the comb were examined. After 96 h, the comb with the second-instar larvae was removed from the hive and returned to the laboratory for larval grafting. The larval diet was prepared according to the protocol developed by Aupinel et al. [65]. The larval diet consisted of 6 g d-glucose, 6 g d-fructose, 1 g yeast extract, 50 g royal jelly, and 37 mL double-distilled water (ddh_2_O). All the materials were weighed one by one, poured into a 100 mL beaker containing 37 mL of ddh_2_O, and mixed well. Royal jelly was stored at −20 °C and placed in a water bath at +34.5 °C for 10 min, then 50 g of royal jelly was poured off and placed into a 100 mL beaker and some of the nonessential particles were removed. Then, the larval diet was pureed and placed into a 20 mL test tube, and the date was marked. We fed this diet to larvae for three days, and the larval diet was prepared using the same method for subsequent experiments.

### 4.6. Larval Grafting and Inoculation

Before grafting, the larvae were placed in cell culture plates, and the food and plates were prewarmed at ±34 °C for 15 min. Then, 20 µL of food was added to the culture plates, and second instar larvae (within 48 h of hatching) were gently grafted by Chinese grafting tools [66]. The grafting tools were placed in ethyl alcohol for 15 min and then washed with distilled water. To prevent cross-contamination by pathogenic microorganisms after grafting 16 larvae, the grafting tools were replaced with a new disinfected tool. Six 48-well cell culture plates (three in the control group and three in the treatment group) were considered for each colony. The culture plates were placed in an incubator (Ningbo Haishu, Saifu Experimental Instrument, Ningbo, China) at ±34 °C. with relative humidity (RH) of 95–99% inside a plastic box saturated with Potassium Sulfate (K_2_SO_4_) 360 g/L water to prevent dehydration. which was prepared in a plastic box. The larvae were fed with a pipette and modified tips, and the remaining food was removed daily. Inoculation tests were conducted based on the method described by Hu et al. [43]. The 3rd instars larvae with a single dose of the virus preparation (Fuzhou SBV at 6.32 × 10^6^ copies/1 µL of the purified virus) were mixed well with 10 µL of food and placed into cell culture plates using a micropipette with a modified tip. The larvae were then allowed to thoroughly consume the food. After 24 h, in treatment groups 1, 2, and 3 and control groups 4, 5, and 6, no virus was added to the diet in all subsequent portions of the larval food. At 7 days, the larvae began defecating, determined by glycol uric acid crystals and fibrous substances on the well. The larvae were gently transferred by plastic forceps to the new cell culture plates, which were lined with two layers of sterilized filter paper (autoclaved and dried). Culture plate larvae were cleaned with sterilized tissue paper before being transferring to new cells. The plates were placed in incubators at ±34 °C and 75–80% RH (360 g/L saturated salt water) until pupation. The larvae were reared for an additional six days.

### 4.7. Observation and Sample Collections

Mortality was recorded daily, and larvae were observed under a microscope for immobility, swelling, tissue breakdown, and infection with SBV. The larvae that showed symptoms of SBV were individually collected in a 1.5 mL Eppendorf tube and marked. When the larvae showed discontinued respiration and reduced flexibility, developed edema, and exhibited a color change to brownish, they were removed from the cell culture plates every day and recorded. Therefore, they could be contaminated and decomposed by bacteria and fungi. Larvae infected with SBV were distinguished, collected during observation, recorded, and stored at −80 °C for subsequent experiments [67].

### 4.8. Sample Preparation and DNA Isolation

After four months of larval rearing, we collected a sufficient number of samples. We randomly selected 90 samples (45 S at the 7–10th instar larva stage and 45 R at the 12–14th pupa stage) from three *A. c. cerana* colonies with three biological replicates. Each colony was composed of 15 R and 15 S for genomic DNA extraction. Genomic DNA was extracted from whole larvae and pupae using the Cetyl Trimethyl Ammonium Bromide (CTAB) method of Paterson et al. [68]. The samples were ground into a powder with liquid nitrogen and quickly transferred into a 1.5 mL Ep tube, where 800 μL of CTAB extraction DNA buffer was added and mixed well. CTAB was preheated at 65 °C in a water bath, gently oscillated several times every 5 min, centrifuged at 12,000 rpm for 20 min, carefully centrifuged for 15 min, and the supernatant was transferred to a fresh Eppendorf tube. An equal volume of phenol-chloroform 800 µL solution was added, well mixed, and then centrifuged at 12,000 rpm for 10 min at 4 °C. The supernatant was carefully pipetted, and an equal volume of chloroform was added and centrifuged at 12,000 rpm for 10 min at 4 °C. Step 4 was repeated twice. The supernatant was precipitated at −20 °C for 1 h and then centrifuged at 12,000 rpm for 10 min at 4 °C. The supernatant was discarded and the pellet was washed twice with 70% ethanol. After drying at room temperature (generally 5–15 min), the samples were dissolved in 30 μL DEPC water and stored at −20 °C for later use. The concentration and quality of DNA were measured by measuring its absorbance at OD _260/288-_ OD _260/230_ nm using a Nanodrop (Thermo Scientific _2000_/_2000_ Waltham, MA, USA). A spectrophotometer was used to measure 1 µL of the DNA sample, and it was recorded directly. The DNA quality was visually monitored by electrophoresis on 1.0% agarose gel. Qubit 4.0 (Thermo Fisher Scientific, Waltham, MA, USA) was used to quantify DNA using a fluorometer and diluted to 50 ng/µL for sequencing. The amount of DNA used for whole-genome sequencing was 2 µg per sample. For SNP validation, larvae were reared using the same method described previously. The samples were removed at −80 °C to a refrigerator (Haier, Kyoto, Japan) and dissected with sterilized scissors. The thorax from the pupae and half of the larvae were used for DNA extraction.

### 4.9. Sample Sequencing and Data Processing

Data sequencing and bioinformatics analyses were conducted based on the pipeline. The experiments were conducted according to the standard procedure developed by Illumina (San Diego, CA, USA). Qualified DNA samples were resequenced by whole-genome sequencing using a commercial Illumina NOVA HisSeq X-Ten Biomarker Technologies Corporation (Beijing, China). Sample quality detection, genomic DNA library construction, library quality detection, and sequencing were also included. After the genomic DNA test, qualified DNA samples were fragmented by mechanical interruption (ultrasound), fragmented purification, end pair, 3′ ends plus A, and connection sequencing joints; electrophoresis on 1.0% agarose gels was used to select fragment sizes. Polymerase chain reaction (PCR) amplification was performed to construct a sequencing library. DNA libraries were constructed using the DNA Library Prep Reference Guide with an insert size of 150 bp and were constructed based on Illumina, following the manufacturer’s instructions (Illumina Genome Analyzer II), and paired-end reads were generated using Illumina. The DNA library was first built to conduct a quality inspection, and a qualified library was used for sequencing. The quality of the first reads (double-end sequence) was obtained by sequencing, evaluation, and filtering. Clean reads were used for subsequent bioinformatics analyses. Clean reads were compared with reference genome sequences, and variations in SNPs, indels, and other variants were detected and annotated based on the comparison results. Subsequent genetic evolution analyses were performed based on the detection of variation.

### 4.10. Sequence Data Filter, Quality Control, and Data Assessment

Quality control (QC) processed low-quality data by filtering the sequenced reads or raw reads (Appendix A). To ensure data quality, raw reads were filtered and clean reads were obtained and used for subsequent bioinformatics analyses before aligning them to the reference genomes of *A. c. cerana*. Low-quality reads containing adaptor sequences, duplicate reads, and reads containing adaptor sequences were filtered based on the following three rules: the main steps of data filtering were the removal of adapter reads, removal of >10% Ns reads, and removal of >50% reads with low bases (base quality scores <10). The raw data or raw reads and the results were in the FASTQ format. They were stored in fq file format, containing sequence information for the sequencing reads and corresponding sequencing quality information. Clean sequencing data were used for SNP calling.

### 4.11. Mapping Variant Detection and SNP Calling

The clean reads were calculated using the SAMTOOLS flag stat command [69]. Filtered clean reads were compared with reference genome sequence reads that were mapped to the reference genome of *A. c. cerana* (*APICC*1.0, GenBank, Assembly GCA_002290385.1) using the ‘mem’ algorithm of Burrows–Wheeler Aligner (BWA version 0.7.5a-r405) [44], with the default parameters. The mapping results were processed by sorting and duplicate marking using SAMTOOLS (version 0.1.19–44428 cd), and variants were detected [69]. Clean data were used to call both SNPs and Indels using PICARD tools (http://broadinstitute.github.io/picard/ (accessed on 26 December 2018) and local realignment around Indels. The Indels-Realigner in GATK Haplotype Caller algorithm in GATK software version 3.8 was used for SNP SNPs and small Indels mutations calling across the 90 samples with default parameters [40]. Raw variances, including SNPs and indels with low quality (QUAL < 30, QD < 2.0, FS > 60.0, MQ < 40.0), were filtered. Distribution of base sequencing quality (Base-calling analysis, Illumina Casava Software version 1.8) was used with sequencing parameters (paired-end PE; sequenced read length, 150 bp). The average sequencing depth was 14x coverage. Low-quality reads were filtered based on the following four rules: If one end of a pair-end read had > 5% Ns bases, the paired-end read was removed. Each pair-end read was removed if it had an average base quality of Q < 20 (Phred-like score). Each read was trimmed to its three bases if their quality scores were Q < 13. Trimming was stopped at the base with a quality score of ≥ 13. If the number of remaining bases was <40, then the paired-end reads were removed. Duplicates of paired-end reads were removed.

### 4.12. Variation Detection, Quality Control, and Annotation

SNPs and small Indels were detected using Genome Analysis Toolkit (GATK) software [40]. Based on the positioning results of the clean reads in the reference genome, PICARD tools (Picard: http://sourceforge.net/projects/picard/ (accessed on 26 December 2018) (Picard) were used to filter redundant reads (mark duplicates) to ensure the accuracy of the test results. The variation detection of SNPs and indels was carried out using the GATK haplotype caller (local monomer assembly) algorithm, and each sample was first generated in genome variant call format (gVCF). Then, the group joint-genotyped all VCF SNPs (Appendix A). Finally, the filter was set on the final mutation sites using the variant call format (VCF) in the R private SNP VCF R private SNPs. The annotation line contained the message of the file data row and the interpretation of the meaning of the various identifiers was used in the format column. In contrast, the header and data row contain the variation detection results of each sample. The variation results were strictly filtered to ensure reliability; the main filtration parameters were as follows: based on the subroutine in BCF tools, vcfutils. Pl (var filter-w 5-w 10), the SNP and adjacent Indels in 5 bp near Indels were filtered out within 10 bp; cluster size 2 cluster window size 5 bp, indicating that the number of variations in the 5 bp window should not be exceeded. The quality value of the Q < 30 Phred format demonstrates that there is a possibility of divergence at that point. Suppose that the mass value is >30, which was filtered out. In this case, QD < 2.0, the variant mass value (quality) was divided by the ratio of coverage depth, and the covering depth was the sum of the depth of coverage of all samples containing mutant bases at this point. QD < 2.0 were filtered out. If the MQ was <40, all ratios were read on the bit point to the mean square root of the mass value. If the MQ was >40, it was filtered out. If the FS was > 60, the value was converted through the *p* value < 0.05 of the Fisher exact test, which described whether there was a significant positive or negative chain specificity for the reads that contained only variations and reads that had only reference sequence bases during sequencing or comparison. There are no chain-specific comparison results, and the FS should be close to zero. An FS above 60 was filtered out, and other variation filtration parameters were processed using the default values specified by GATK.

### 4.13. Annotation of Mutated Genes at the DNA Level

Annotation was conducted to classify SNPs and Indels as genetic variants using the SnpEff toolbox [70], which was used to annotate the effect impact, function class, codon change distance, amino acid change, gene name, transcript biotype, gene coding, transcript ID, exon, and intron rank warning error for each SNP. An annotation analysis of differentially expressed genes was the basis for further explanation of gene function. The KEGG database is the primary public database for this pathway. The presence of genes in specific regions was studied using pathway enrichment analysis [71].

### 4.14. Population Genetics Analysis

To study the relationships and evolution between SNPs and traits, a phylogenetic analysis of all detected SNP genotypes after filtering was used to construct the phylogenetic tree SNPs with low MAF (<0.05). A low genotyping rate (*p* ≤ 0.5) was used from 90 samples using the neighbor-joining (N-J) method under the Kimura 2-parameter model implemented by M_EGA_ X Software, with bootstrap replicates of 1000 [72]. In addition, to study the relationship and evolution between R and S in SNP validation, phylogenetic trees were constructed from R and S from different SNP validation data. Phylogenetic tree visualization and editing assignments were performed using ITOL (http://itol.embl.de/ (accessed on 13 March 2019).

The EIGENSOFT software package was used to conduct PCAs based on SNP data characteristics to obtain sample clustering. The PCs were used to visualize the group of samples, which could explain the percentage of the total variation based on SNP data to carry out PCA and obtain the clustering of samples [73]. An admixture analysis in the haploid model, Admixture 1.3, was used to filter SNPs with multiple K values (i.e., the number of putative populations) ranging from 1 to 10, and the CV error was used to select the best K-value [74]. SPAGeDi (Spatial Pattern Analysis of Genetic Diversity) software was used to estimate the relative kinship between two individuals in a natural population [75]. 

### 4.15. Identification of Selective Sweep-between Traits

The F_ST_ and π ratios were calculated to detect differentiation and reveal the significant regions using PopGenome [41]. To detect genomic regions that were potentially differentiated for R and S, the F_ST_ and π rates in the top 1% and top 5% were calculated for 100-kb sliding windows with a step size of 10 kb using VCFTOOLS. Sliding windows with F_ST_ values of > 95% or > 99% and π ratios of > 95% or > 99% of genome-wide F_ST_ values were selected and regarded as significantly different windows. Overlapping significance windows were merged into single fragments, and the fragments were regarded as highly divergent regions across groups. To identify genes putatively under selective sweep between the two populations, SnpSift software was used to screen each SNP site. Based on the SNP results of the mutation detection, filtering, and selective sweep, an INT of < 0.5 and an MAF of < 0.05 were used to obtain highly consistent SNP loci in the genome-wide empirical distributions. F_ST_: Estimation of gene flow levels from the DNA sequence data. π: Statistical test for detecting geographic subdivisions [45].

### 4.16. Functional Annotation and Enrichment Analysis

Functional and annotation analyses of 90 individual samples were conducted, and their gene changes were detected and annotated against ten databases. After detecting the significant regions by selective sweep between traits, the genes and SNPs were annotated. The enrichment factor was used to analyze the enrichment degree of the pathways, and Fisher’s exact test was performed and the false discovery rate (FDR) was determined to calculate the significance of enrichment with a *p* value of < 0.05.

### 4.17. SNP Validation and Sanger Sequencing

To verify the candidate SNPs, we first conducted an (MAF < 0.05) analysis to select high-consistency SNPs based on the combination of F_ST_ and π S divided R top 1% and F_ST_ and π R divided S top 1% with a high percentage genotype. For significant SNPs, 40 pairwise primers were designed to amplify the desired SNP for SNP validation (Appendix A). In the PCR, the total reaction volume was 25 μL, containing the following components: 2 μL DNA template, 0.4 μL of each primer (forward and reverse), 9.7 μL ddH2O, and 12.5 μL 2X Fine Easy Taq Super Mix Trans solution (TransGene, Beijing China). The PCR amplification was conducted at 94 °C for 3 min, followed by 35 cycles of 94 °C for 30 s, a temperature of between 57 °C and 58 °C for 30 s, 72 °C for 1 min, and a final step at 72 °C for 5 min. To ensure that the PCRs were running correctly and to prevent contamination, the *Apis*-β-actin primer pair (181 bp) was used as an internal positive control [76]. A negative control, ddH_2_O, was used for all PCR amplification assays. Amplified PCR products were analyzed using 5 μL of 2% agarose gel electrophoresis and visualized under UV light. The lengths were compared to that of a standard Trans 2 K molecular weight DNA ladder (Invitrogen, Waltham, MA, USA). After visualization, the PCR product of 20 μL of amplified target DNA was purified, positive strands (forward primer) were used in di-deoxy Sanger Sequencing, and the ABI file data were analyzed.

### 4.18. Statistical Analysis

GraphPad Prism version 9.00 (121) for Windows (Graph Pad Software, San Diego, CA, USA) was used to compare the control group with the treatment group to evaluate the survival function analysis. The Kaplan–Meier chi-square log-rank (Mantel–Cox) test was performed, where a *p* value of < 0.01 was considered significantly different. Additionally, the survival functions of CR and CS were compared. For SNP flanking validation, specific primer pairs were designed using the default parameters of the BatchPrimer3 v1.0 online software. Chromas, BioEdit sequence Alignment Editor, and DNASTAR Lasergen SeqMan Pro were used to manually evaluate each sample peak and compare the variance in the genotype differences. The distributions of allele frequencies between R and S were analyzed using the chi-square test in Hardy–Weinberg equilibrium, where a *p* value of < 0.05 was considered significantly different. The ggplot2 package in R was used to plot the significant region graphs.

## 5. Conclusions

SBV is prevalent in the genus *Apis*, including bumblebees, and effective methods to control this viral disease are urgently needed. Genome sequences are crucial for basic biological research on bees, and Asian honeybees provide an excellent model for understanding the genetic basis of SBV resistance at the genomic level. To catalog the genetic variations and nucleotide polymorphisms associated with SBV resistance in the *A. c. cerana* genome, bee larvae were challenged with SBV and raised in vitro, and 90 individual samples from three populations were resequenced. This study identified 31,000,613 high-quality SNPs with 696,352 individual genotypes compared to the reference genome. Two SNP molecular markers related to SBV resistance were identified and successfully validated by multiple verifications. The results showed that the two SNPs were significantly associated with SBV resistance, and these significant SNP loci were found on the 15th chromosome, composed of five putative candidate genes. The favorable G/G and C/C allele frequencies in the R samples at the two SNP loci revealed the molecular mechanism of SBV resistance by validation and analysis of the allele frequencies. Functional and enrichment analyses of predicted genes showed that genes involved in cellular processes might play a significant role in SBV resistance. The findings of this study may support and improve the honeybee *A. c. cerana* genome and may be employed in breeding to enhance SBV resistance and decrease SBV infection. Hence, SNP molecular markers can quickly and accurately identify SBV-resistant colonies, and further investigation is needed to determine if the two discovered SNPs are also responsive to other closely related species, such as *A. mellifera*.

## 6. Patents

This study received two patents from the China Innovation Center which have already been published. Patent. Issue No. 2021030301132160. Publication No. CN 112430675 A, SNP labeling at KZ288474.1_32271. Another Invention Issue No. 2021100418633. Publication No. CN 112430675 A, SNP labeling at KZ288479.1_95621. Invention method for identifying SNP molecular markers related to sacbrood virus resistance.

## Figures and Tables

**Figure 1 ijms-24-06238-f001:**
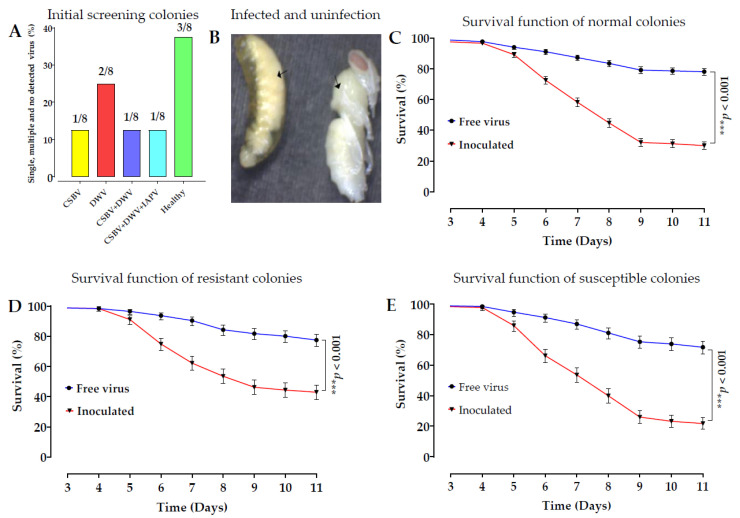
Screening colonies, infected and uninfected larva, and probability survival function. (**A**) Screening candidate colonies against seven common bee viruses. Based on RT–PCR detection of viruses, three out of eight colonies were infected with a single virus (CSBV and DWV), two were infected with multiple viruses (CSBV, DWV, and IAPV) in the three colonies, and no virus was detected. (**B**) Infected larvae on the left ← and resistant larvae (pupae) on the right side → after inoculation with SBV. (**C**) To assess the efficient infectiousness of the purified virus, a comparison of the survival function of normal colonies with the free virus group with the treatment groups was conducted using a log-rank (Mantel–Cox) test with 95% CI ratio, where a *p* value of < 0.01 was defined as significantly different. ꭓ^2^ = 595, df = 1, *n* = 3 colonies. Each colony comprised 144 larvae with three replicates of the control and treatment groups of nine months. A total of 1296 samples from two groups were included in the data analysis. (**D**) Comparison of the survival function of the colony-resistant control with that of the treatment groups. ꭓ^2^ = 115.2, df =1, *n* = 3 colonies. A total of 864 samples with three replicates from the control and treatment groups were included in the data analysis. (**E**) Comparison of the survival function of susceptible colonies control with the treatment groups. ꭓ^2^ = 229.5, df = 1, *n* = 3 colonies. The data from these comparisons between the control and treatment groups, and between resistant and susceptible colonies and with the control groups, also showed significant differences (*p* < 0.001). However, comparison analyses of the control groups with the control groups of all colonies were not significantly different (*p* > 0.01).

**Figure 2 ijms-24-06238-f002:**
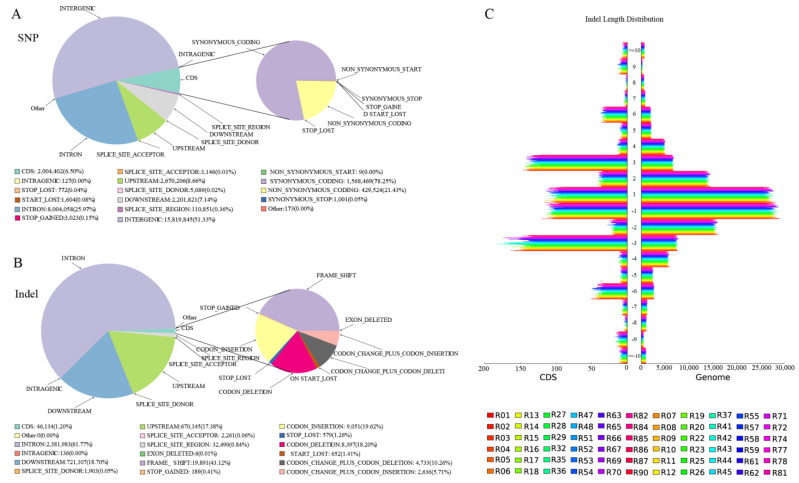
Genome variations, annotation, and indel length distribution map in CDS, condign area, and the whole genome. (**A**) A plot of all SNP annotations. (**B**) A plot of all indel annotation distributions. (**C**) Indel length distribution in CDS and the genome. The longitudinal *Y*-axis coordinates are indel lengths (within 10 bp); greater than 0 is an insertion and less than 0 is a deletion. The horizontal *X*-axis coordinates are the corresponding quantities. R represents resistance and S represents the susceptibility of 90 samples in the three populations of *A. c. cerana*.

**Figure 3 ijms-24-06238-f003:**
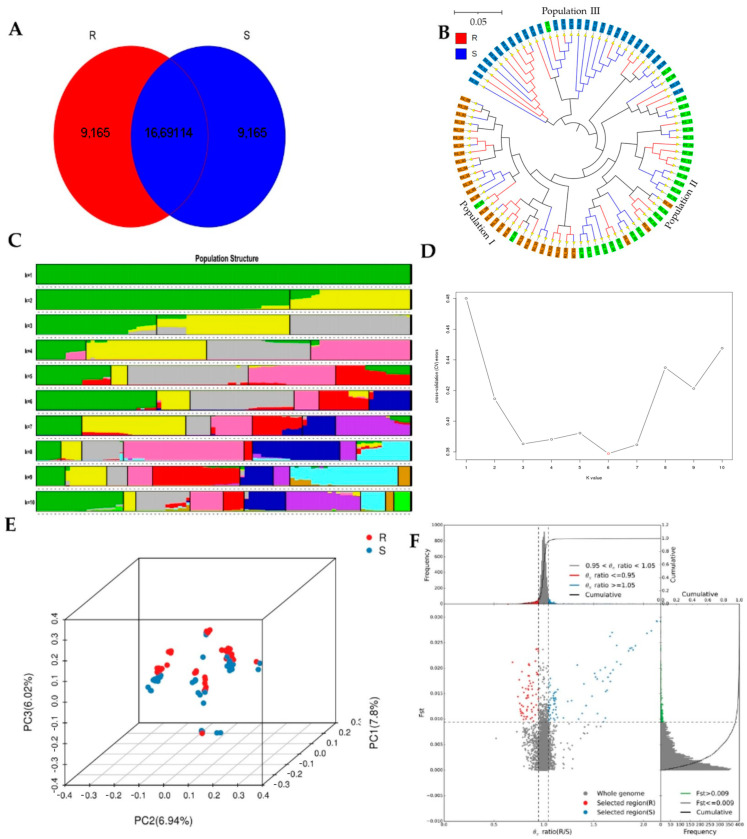
Venn diagram of unique SNPs, phylogenetic tree, population structure, principle component analysis, and selection signature for resistance to SBV. (**A**) Venn diagram of unique SNPs and shared SNP genotypes between R and S. By using 1,797,078 VCFs and considering the 50% optimal genotype in the R group, we obtained 9165 unique SNPs in R and S, and 669,114 SNP genotypes were not found to be the cause of 118,801 SNP loci in at least one subgroup that was under 50% optimal genotypes. Such loci were not involved in analyzing the group. (**B**) A phylogenetic neighbor-joining tree was constructed using 696,352 SNPs with 1000 bootstrap replicates of three populations. (**C**) Population structure and admixture of 90 samples between R and S from three populations and the cross-validation error rate. The *x*-axis represents the K values from 1 to 10 and the *y*-axis represents the cross-validation (CV) errors between 0.38 and 0.48, where the best K was 6. The accessions were divided into ten subgroups (there was a minimum K-value when K = 6); within each subgroup, the accessions were ordered according to the genetic component, and each line gives the subgroup value. Each accession shown as a vertical line partitioned into K colored components represents inferred membership in K genetic clusters. (**D**) A 3D principle component analysis (PCA) plot was constructed using SNP genotypes of the two groups (R and S) with a 3D plot. (**E**) Distribution of selective sweep and population differenciaciation index (F_ST_) with nucleotide polymorphism (θ_π_) ratio selection regions of genes and SNPs for the top 5% R (resistant) and S (susceptible) *A. c. cerana* larvae and (**F**) The horizontal coordinate represents the ratio of θ_π_ (π ratio); the vertical coordinate represents the F_ST_ value, corresponding to the frequency distribution diagram above and the frequency distribution diagram on the right, respectively; and the dot plot in the middle represents the corresponding ratio of F_ST_ to θ_π_ in different windows. Gray represents the whole genome; the red and blue areas at the top are the top 5% regions selected by θ_π_, the green area is the top 5% region selected by F_ST_, and the blue and red areas in the middle are the intersection of F_ST_ and θ_π_, which are the candidate sites. The distributions of threshold line F_ST_ < 0.95, θπ < 1.05, F_ST_, and θπ < 0.95 ratios of the top 5% strong signal region analysis on genome selection of genes were calculated in 10 kb windows sliding in 1 kb steps in R and S, respectively.

**Figure 4 ijms-24-06238-f004:**
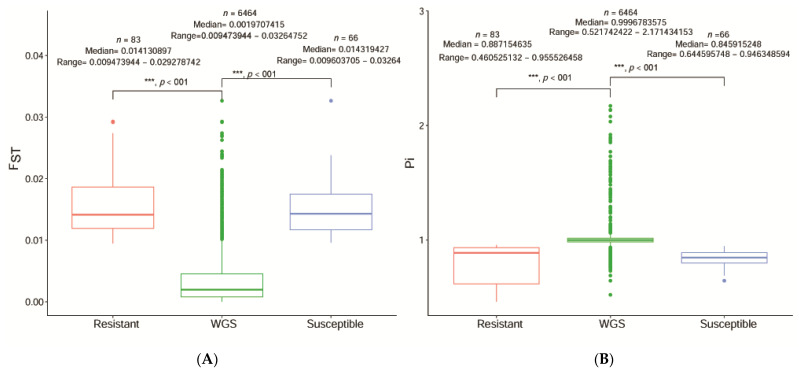
Distributions of selective sweep analysis results of significant regions of intersection F_ST_ and π ratios between R and S in *A. c. cerana* larvae that had undergone positive selection the top 5% of the whole genome. (**A**,**B**) Comparison analysis of selected regions based on population differentiation index (F_ST_) and nucleotide polymorphisms (π) of the significant intersecting regions (top 5%) of genome variants between R (resistant) larvae and S (susceptible) larvae of *A. c. cerana*. Data were reanalyzed and plotted from analyzed significance regions.

**Figure 5 ijms-24-06238-f005:**
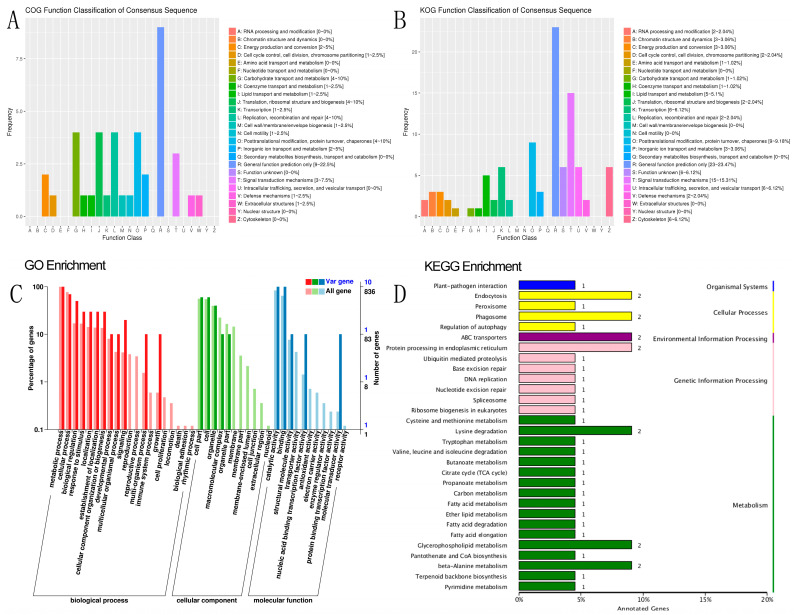
(**A**) Clusters of Orthologous Genes (COG); (**B**) Gene Ontology (GO); (**C**) KOG (Eukaryotic Orthologous Groups)), and (**D**) Kyoto Encyclopedia of Genes and Genomes (KEGG) of top 5% population differentiation index (F_ST_) and nucleotide polymorphisms (π) in *A. c. cerana*.

**Table 1 ijms-24-06238-t001:** An overview of the sequenced data, identified SNPs, indels, and annotations of 90 samples in *A. c. cerana*.

Data Sequences		SNP and Indel Detection		Annotated SNP, Indels, and Genes	
Base sum	3.37 × 10^11^	Raw filter SNP	31,000,613	Raw SNP annotation VCF	1,797,078
Read sum	1,126,922,588	Transition	25,586,296	Filtered SNP annotation GATK	1,048,575
Total reads	2,253,845,176	Trans-version	5,414,317	Indels genotype VCF	424,131
Raw reads	1,127,774,116	Heterozygosity	18,163,8808	Raw filter Indels annotation GATK	424,131
Clean reads	1,126,922,588	Homozygosity	12,836,805	Gene with Indels	91,114
Ave. GC	34.16%	SNP genotype info cloud	872,228	Genes with nonsynonymous SNP	308,336
Ave. coverage 14X	99.04%	DEG SNP	1,769,990	Annotated genes of 90 samples	9359
Ave. mapped	87.92%	Final SNP genotype	696,352	300 < = length < 1000	2997
Ave. properly mapped	82%	Private S SNPs	286,552	Length > = 1000	6135
Genome: total	11,036,568	Private R SNPs	275,422	Significance regions entire genome at scaffold level	457

VCF = Variant Call Format, GATK, Genome Analysis Toolkit.

**Table 2 ijms-24-06238-t002:** Distributions of G and C allele frequencies of SNP KZ288474.1_322717.

Category	Colony	Sample size	Frequency of genotypes			Frequency of alleles^c^						
			G/G	G/C	C/C	G	*P_G_ * ^a^	Mean ± SD	C	*Pc* ^b^	Mean ± SD	Remark
	R1	48	26	20	2	72	0.750		24	0.250		First and second validation
R	R2	48	37	11	0	85	0.885	0.843±0.066	11	0.115	0.156±0.066	
	R3	48	38	10	0	86	0.896		10	0.104		
	S1	48	4	40	4	48	0.500		48	0.500		
S	S2	48	8	40	0	56	0.583	0.580±0.064	40	0.417	0.420±0.064	
	S3	48	15	33	0	63	0.656		33	0.344		
	CR1	64	53	11	0	117	0.914		11	0.086		Comparative validation
CR	CR2	64	48	16	0	112	0.875	0.898±0.017	16	0.125	0.102±0.017	
	CR3	64	52	12	0	116	0.906		12	0.094		
	CS1	64	28	31	5	87	0.680		41	0.320		
CS	CS2	64	36	20	8	92	0.719	0.685±0.026	36	0.281	0.315±0.026	
	CS3	64	25	34	5	84	0.656		44	0.344		
	R1	162	104	54	4	262	0.804		62	0.196		Further validation
R	R2	32	26	6	0	58	0.906	0.862±0.043	6	0.094		
	R3	32	25	6	1	56	0.875		8	0.125	0.138±0.043	
	S1	112	45	45	22	135	0.603	0.648±0.059	89	0.397		
S	S2	32	12	14	6	38	0.731		26	0.269	0.352±0.059	
	S3	32	9	21	2	39	0.609		25	0.391		

R, resistant S, susceptible CR, resistant colony CS, susceptible colony, ^a^ frequency of the G allele (*P_G_*),**^b^**frequency of the C allele (*P*_C_), and ^c^ the differences between C and T allele frequencies and colony category in exact Chi-square test at a *p*-value of <0.05 was defined as a significant difference. The overall results showed a significant difference between the resistant allele (G) in R and susceptible allele (C) in S, ꭓ^2^ = 15.595, df = 1, *p* = < 0.000078.

**Table 3 ijms-24-06238-t003:** Distributions of C and T allele frequencies of SNP KZ288479.1_95621.

Category	Colony	Sample Size	Frequency of Genotypes			Frequency of Alleles ^c^						
			C/C	C/T	T/T	C	*P* _C_ ^a^	Mean ± SD	T	*P_T_ * ^b^	Mean ± SD	Remark
R	R1	48	25	23	0	73	0.760		23	0.240		First validation and second validation
	R2	48	22	26	0	70	0.729		26	0.271		
	R3	48	37	11	0	85	0.885	0.792 ± 0.068	11	0.115	0.208 ± 0.068	
S	S1	48	4	38	6	46	0.479		50	0.521		
	S2	48	4	40	4	48	0.50	0.542 ± 0.074	48	0.50	0.458 ± 0.074	
	S3	48	14	34	0	62	0.646		34	0.354		
CR	CR1	64	46	18	0	110	0.859		18	0.141		Comparative validation
	CR2	64	38	24	2	100	0.781	0.757 ± 0.094	28	0.219	0.242 ± 0.093	
	CR3	64	23	35	6	81	0.633		47	0.367		
CS	CS1	64	35	18	11	88	0.688	0.614 ± 0.071	40	0.313	0.385 ± 0.077	
	CS2	64	28	27	9	83	0.648		45	0.352		
	CS3	64	1	63	0	65	0.508		63	0.492		
R	R1	162	113	47	2	273	0.843		51	0.157		Further validation
	R2	32	8	16	8	32	0.50		32	0.50		
	R3	32	25	7	0	57	0.891		7	0.109		
	R4	32	21	10	2	52	0.797	0781 ± 0144	14	0.203	0.219 ± 0144	
	R5	16	13	2	1	28	0.875		4	0.125		
S	S1	112	50	41	21	141	0.629		83	0.371		
	S2	32	5	17	10	27	0.422		37	0.578		
	S3	32	15	17	0	47	0.734	0.570 ± 0.105	17	0.266	0.430 ± 0.105	
	S4	32	8	18	6	34	0.531		30	0.469		
	S5	16	4	9	3	17	0.531		15	0.469		

R, resistant S, susceptible CR, resistant colony, CS, susceptible colony, **^a^**frequency of C allele (*P*_C_), **^b^** frequency of T allele (*P_T_*), and ^c^ the differences between C and T allele frequencies and colony category in exact Chi-square test at a *p*-value of < 0.05 was defined as a significant difference. The overall results showed a significant difference between the resistant allele (C) in R and the susceptible allele (T) in S (ꭓ^2^ = 8.931, df = 1, *p* < 0.0028).

## Data Availability

Data from 90 sequenced individuals were deposited in the Sequence Read Archive (SRA) of the National Center for Biotechnology Information (NCBI) under the accession number BioProject: PRJNA815733.

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
