# Peer review of "Discovery of SNP Molecular Markers and Candidate Genes Associated with Sacbrood Virus Resistance in Apis cerana cerana Larvae by Whole-Genome Resequencing"

_ijms, 2023, doi:10.3390/ijms24076238_

Round 1

Reviewer 1 Report

In this manuscript, the authors try to locate SNPs associated with SBV resistance in honey bees, to investigate the variations in the genomic DNA between R and S larvae using whole-genome resequencing and annotation of these results and to identify SNP regions, genotypes, and associated genes. Overall, this is an interesting manuscript, appropriate for publication in the IJMS, but I feel that introduction section is very lengthy and results sections need to be revised significantly in accordance to supplementary results. I strongly encourage the authors to dedicate the time and improve the quality of their manuscript.

Minor comments:

-Tables and figure legends needs to be arranged in order.

-The English language also needs to be improved throughout the manuscript, but does not detract from the science. The authors are advised to seek the assistance of a native English speaking colleague.

Author Response

Open Review

In this manuscript, the authors try to locate SNPs associated with SBV resistance in honey bees, to investigate the variations in the genomic DNA between R and S larvae using whole-genome resequencing and annotation of these results and to identify SNP regions, genotypes, and associated genes. Overall, this is an interesting manuscript, appropriate for publication in the IJMS, but I feel that introduction section is very lengthy, and results sections need to be revised significantly in accordance to supplementary results. I strongly encourage the authors to dedicate the time and improve the quality of their manuscript.

Minor comments:

-Tables and figure legends needs to be arranged in order.

-The English language also needs to be improved throughout the manuscript but does not detract from the science. The authors are advised to seek the assistance of a native English speaking colleague.

Response: Thank you so much for your interest in our work; we really appreciate your constructive and critical comments.

The introduction became shorter with considering the flow. Some parts of the results, such as PCA, Venn diagram, phylogenetic tree, and population structure analysis, were added to the main body; files from supplementary were moved to the main body, and the analysis description was revised. Materials and methods became a little bit shorter and have been fixed.

Response: Now we addressed the figures and tables in order, and may you see the changes in the revised version.

Response: Once again, thank you for your suggestions. We have applied for English language editing we seriously asked them to scientifically and comprehensively improve the English language; they have already conducted revisions we hope to be more precise and readable than before.

Reviewer 2 Report

Discovery of SNP Molecular Markers and Candidate Genes Associated with Sacbrood Virus Resistance in Apis cerana cerana Larvae by Whole-Genome Resequencing Based Selective Sweep  Analysis

The submitted paper is a important contribution. The data is excellent. The conclusions are clear and valuable.  The paper should be published.  However, there is a good bit of work needed to make the paper readable.

 Reading the manuscript is a challenge. Occasionally the wording is entertaining, viz. A reference to hens that appears to be a reference to queen bees. At other times, a word is inserted at the first of a sentence that makes the sentence unreadable.  Deleting the word is usually all that is needed.  Often the verb is left out and the sentence is a fragment. When I can, I suggest the omitted verb; too often I can’t think of a verb that makes sense of the fragment. In those cases, the fragment should be integrated into the proceeding or following sentence.

Occasionally, a sentence will have with multiple verbs.  Multiple verbs works in a compound sentence, but render a simple sentence unreadable. I do what I can to suggest the  verb to delete.

The paper is complex. Every analysis includes 8-10 summary statistics including Fst and Pi, plus others summary statistics I know less well. It is never clear how this overkill helps.

A few figures show graphs that look different but are captioned the same. This does not work..

Anytime I read the phrase “statistical analysis shows” I have to read and re-read. Sometimes this phrase refers to counts. Often I could not determine what it references. Reading the Methods section, the reader learns what the phrase means. However,. I would delete this phrase everywhere it is written. The deletion will necessitate splitting the methods up as described below.

A related problem to the above is “additional statistical analysis”? This last may refers to a chi-square with data from the 5 comparisons summed together. The suggestion regarding splitting up the methods will also address this problem phrase. It tells the reader nothing and should  be deleted.

The discussion is quite good.  It is focused and readable.  It is notable that the discussion works without reference to the many summary statistics analyzed.  I have to wonder why the results are complicated by multiple comparisons based on different summary statistic. My best guess is that each of the summary statistics is a favorite of one of the authors.  If so, the same authors have been remiss in not discussing what each of the different summary statistics contributes to the take-home message of the paper. If that boils down to the message that all the summary statistics tell the same story, that is a valuable observation and should be clearly stated.

The material and methods reads really well and is very clear. I suggest how and why  the Methods section  should be reduced in length below.

The manuscript will be greatly improved if redundancy (repetition of sentences and portions of sentence) is eliminated. In particular, sentences and phrases from the discussion tht are repeated in other sections should be deleted from tthe manuscript. 

The methods section includes what was done and why. That Needs to be split up. The methods should say what was done. The results should introduce why it was done and report what was found. The latter is currently missing or is so poorly described that the reader can make no sense of the results until learning the why in methods. Where the explanation “why is  needed is often signaled by  the Phrase “Statistical analysis shows” .

The problem of what and why extends to how many colonies, what is meant by validate, and why different locations are mentioned .I am hopeful these problems can be addressed, but do not try to do so here.

I like to see the use of PCA and LD.  However, the results from Principal component analysis and the Calculations of LD are not reported in results. They should be.  The discussion of these is either absent or a cut and paste from  methods. It is not clear what the reader will learn on seeing  the PCA or the LD figures and data in a supplemental figure. That should be made clear. 

Specific suggestions follow:

Line

46-47. Sentence fragment. No verb. Correct as in 48,49.

54. Delete “although”. Otherwise, this is again a sentence fragment.

57 Delete “. This method is employed”. The resulting single longer sentence will make more sense to the reader.

2, 58 Move the sentence to after citation 12 line 51.

1, 75. Hens?  Rewrite as queens?

1, 82-83 Two verbs, but not a compound sentence. Delete “has”.

1, 87. In 1% of the populations makes no sense, nor does 1% frequency.  Perhaps what is meant is that SNPs represent 1% of the sequence? Alternatively “SNPs” occur at 1% frequency among populations (of honey bees?). Reference?  Is the SNP average frequency really that low?  

1, 97. This makes no sense.  GWAS is a method, SSRs are sequence. It may be  the intension is to point out the use of SNP in GWAs and then point out that  SNP variants are common in and markers of SSR sequences. It would be correct to rewrite that SNPs and SSRs are useful markers in GWAS.

2, 112. Paragraph structure is a major problem.  This is easiest to address using a key sentence to introduce the content of each paragraph. In scientific writing, the Key sentence is usually the first, but can be the second sentence in the paragraph. The rest of the paragraph should support the statement in the key sentence.

Line 112, if moved to the first sentence Line 112 would make a good Key sentence for this paragraph. Line 84, is not. The paragraph is NOT devoted to evidence that “(SNPs) are more accurate and powerful in numerous perspectives of research. “

Line 117. Delete Nevertheless

159. They? The statistics are not adequately described.  The comparisons in 1 C, 1D, and 1E are not clear.  All have the same captions, Control and SBV. It would really help if the X axis label included the comparisons in each. Control and SBV labels alone are not helpful at all.  

If I read figure 1 correctly, there are an undescribed number of colonies that were assayed for presence or absence of 7 common honeybee viruses (primer sequences not given).  Number pairs above each bar may relate to the number of colonies assayed, but the numbers are hard to read,  not explained, and not helpful. Figure 1A shows that colonies were detected with 1) no virus, 2) SBV virus alone, 3) DWV virus alone, 4) both SBV and 5) DWV, SBV, DWV and IPV.  Figure I(B) compares infected and uninfected (Healthy vs SBV alone). Other virus effects not related to the paper are not shown.  The reason for the assay is to identify healthy colonies and colonies infected only by SBV. Because these are the only colonies studied further, it is important the total number of colonies in the two classes is given in the figure and text.

200-202. Were the re-sequenced colonies those identified in Figure 1(A) as healthy or infected by SBV alone.  What number of each were sequenced? Delete “yielding” Delete or explain what is meant by “clean” data. “Clean is described in methods. Delete from here. A word that is often used to mean “clean” is vetted.  The phrase  “an average ratio of the samples to the reference genome of 87.92%”  makes no sense to me.  What is meant by ratio here?

208 Typo? is rather than as?

209 Delete “according to statistical analysis” The analysis is undescribed as such and probably not a statistical analysis, but rather a numerical analysis. If so, that need not be stated.  Re: Table 1 references. If this comes from the “Statisitcal analysis” mentioned in line 209, that analysis must be described.

218 Delete either “we identified” or “were detected”. This can read “that were detected” but that addition requires a description of method of detection. The pronoun “It” refers to what?  Write out in full

221-222 hererozygotes homozygotes

224.  Hetero ratio should be rewritten as heterozygosity.

226 “Around 2.68% of heterozygosity, SNP” makes no sense.  The comma should be deleted and the phrase rewritten to make the meaning clear.  I read this to mean, The proportion of heterozygous SNPs in the R was 2.68 compared to XXX in the S.

232. Bad English. I read this to mean, The other two possible mutant types did not differ in frequency.

248. Delete “Form”

252 This mutation must be explained, or the sentence deleted.

278 nonsynonymous mutants is a reference to codons, not genes

280 Typo?  Between the reference “and” the sample.

297-8 fragment. No verb. Was based?  Are shown?

308-309. Fragment. No verb and I can’t guess the verb .

645. Delete “before we used”. That word order works in German, but not in English.

667-668 a fragment and  a strange reference to “just now”.  If the period is removed from the fragment and “just now is replaced with “from which", this reads OK, assuming the revised sentence is true.

684. A remarkable observation.  It suggest mutation in transcription factor, rather than an altered protein. A bit more comment on this could be ground breaking.

697-698. Needs to be rewritten as two clear statements.  This appears to say that larvae thought to be susceptible, when  the they mature are found to be  resistant. I do not recall reading that in results and do not know what this contributes to the discussion.

769. Typo? Close? 

808. It would be good to include this in the introduction.  Lacking this, the reader must guess why A. cerana was selected for the study.

814. Themselves?  Is this a reference to Workers?

826 Delete “as needed” which implies not used.  Write ”as in”  this study.

847. “of 90 accessions" is a dangling phrase. It is not clear what the phrase references.

963. Sentence fragment.

967 delete period after dehydration.

1201. Statistical analysis of survival functions. The paper will be much easier to understand if  M& M follows Introduction and Conclusions follow Discussion. The problem is that as  written, the Results too often repeat what was done, while the question of why it was done (that should appear in Results) is imbedded  in methods. The Reader trying to make it through Results has a very incomplete and confusing view of what was done, and no idea of why it was done.

Finally, I strongly suggest that "based on selective sweep analysis" be deleted be deleted from the title. The resequencing is not based on selective sweep analysis and the discovery of markers and candidate genes is not based on more than selective sweep analysis.

Author Response

Open Review

Comments and Suggestions for Authors

Discovery of SNP Molecular Markers and Candidate Genes Associated with Sacbrood Virus Resistance in Apis cerana cerana Larvae by Whole-Genome Re-sequencing Based Selective Sweep  Analysis

 The submitted paper is a important contribution. The data is excellent. The conclusions are clear and valuable.  The paper should be published.  However, there is a good bit of work needed to make the paper readable.

 Reading the manuscript is a challenge. Occasionally the wording is entertaining, viz. A reference to hens that appears to be a reference to queen bees. At other times, a word is inserted at the first of a sentence that makes the sentence unreadable.  Deleting the word is usually all that is needed.  Often the verb is left out and the sentence is a fragment. When I can, I suggest the omitted verb; too often I can’t think of a verb that makes sense of the fragment. In those cases, the fragment should be integrated into the proceeding or following sentence.

Occasionally, a sentence will have with multiple verbs.  Multiple verbs works in a compound sentence, but render a simple sentence unreadable. I do what I can to suggest the  verb to delete.

The paper is complex. Every analysis includes 8-10 summary statistics including Fst and Pi, plus others summary statistics I know less well. It is never clear how this overkill helps.

A few figures show graphs that look different but are captioned the same. This does not work..

Anytime I read the phrase “statistical analysis shows” I have to read and re-read. Sometimes this phrase refers to counts. Often I could not determine what it references. Reading the Methods section, the reader learns what the phrase means. However,. I would delete this phrase everywhere it is written. The deletion will necessitate splitting the methods up as described below.

A related problem to the above is “additional statistical analysis”? This last may refers to a chi-square with data from the 5 comparisons summed together. The suggestion regarding splitting up the methods will also address this problem phrase. It tells the reader nothing and should  be deleted.

The discussion is quite good.  It is focused and readable.  It is notable that the discussion works without reference to the many summary statistics analyzed.  I have to wonder why the results are complicated by multiple comparisons based on different summary statistic. My best guess is that each of the summary statistics is a favorite of one of the authors.  If so, the same authors have been remiss in not discussing what each of the different summary statistics contributes to the take-home message of the paper. If that boils down to the message that all the summary statistics tell the same story, that is a valuable observation and should be clearly stated.

 The material and methods reads really well and is very clear. I suggest how and why  the Methods section should be reduced in length below.

 The manuscript will be greatly improved if redundancy (repetition of sentences and portions of sentence) is eliminated. In particular, sentences and phrases from the discussion tht are repeated in other sections should be deleted from the manuscript. 

The methods section includes what was done and why. That Needs to be split up. The methods should say what was done. The results should introduce why it was done and report what was found. The latter is currently missing or is so poorly described that the reader can make no sense of the results until learning the why in methods. Where the explanation “why is needed is often signaled by the Phrase “Statistical analysis shows” .

The problem of what and why extends to how many colonies, what is meant by validate, and why different locations are mentioned. I am hopeful these problems can be addressed, but do not try to do so here.

I like to see the use of PCA and LD.  However, the results from Principal component analysis and the Calculations of LD are not reported in results. They should be.  The discussion of these is either absent or a cut and paste from  methods. It is not clear what the reader will learn on seeing  the PCA or the LD figures and data in a supplemental figure. That should be made clear. 

Response: Thank you for your valuable information and critical comments.

We agree that deleting the verb, the article or redundancy in the sentences makes the sentence readable and flow.

As you know, better whole genome sequencing of 90 individuals created huge data, and different screening results came out in significance region selection. In the previous submission, we tried to report the result from the top to down and narrowed the range of regions selection. That is why affected the clarity results seem complex. The revised version only addresses the combined Fst and Pi; the other analyses were removed.

For the figures you mentioned, we first started with the selection signature in the entire genome at the chromosome level, each chromosome level, each scaffold, and for R and S pairwise comparisons. Now we just kept the selected regions. The other figures were deleted.

The additional statistical analysis was deleted, the tables table 5 was merged into tables 3 and 4, and the final analysis Chi-square test was kept. We just considered the three categories of SNP validation; we prefer to keep it as three SNP validations; if we merged all the data into two categories, just R and S, the value of that SNP validation would be lost and make no sense. We extended SNP validation in the different geographic locations and colonies to ensure the reliability of each identified SNP, whether the identified SNP works in other honey bee habitats.

Thank you for your suggestion. Some supporting references were added to discussion.

We added the PCA plot, including population structure, Venn diagram, and phylogenetic tree, in the main body. Also, we described the results part, and some parts of the materials and methods related to these results were revised.

We did not do further analysis for LD, and we just compared the two groups (R and S) in each chromosome. To be more specific and not be confused, we removed LD figures from all supplementary parts of this manuscript.

Specific suggestions follow:

Line

46-47. Sentence fragment. No verb. Correct as in 48,49.

Response: Revised.

  1. Delete “although”. Otherwise, this is again a sentence fragment.

Response: Deleted.

57 Delete “. This method is employed”. The resulting single longer sentence will make more sense to the reader.

Response: Deleted.

2, 58 Move the sentence to after citation 12 line 51.

Response: Thank you for your suggestion the sentence was moved as you suggested.

1, 75. Hens?  Rewrite as queens?

Response: Thank you revised.

1, 82-83 Two verbs, but not a compound sentence. Delete “has”.

Response: Deleted.

1, 87. In 1% of the populations makes no sense, nor does 1% frequency.  Perhaps what is meant is that SNPs represent 1% of the sequence? Alternatively “SNPs” occur at 1% frequency among populations (of honey bees?). Reference?  Is the SNP average frequency really that low?  

Response: Thank you for your correction. We intended to point out the DNA variations in human genomes; about 99.9% of the DNA sequence is identical. The remaining 0.1% of DNA contains sequence variations, called SNP or snips, by “0.” This misspelling made the statement ambiguous. Cause of the length introduction, the sentence was deleted.

1, 97. This makes no sense.  GWAS is a method, SSRs are sequences. It may be the intention is to point out the use of SNP in GWAs and then point out that  SNP variants are common in and markers of SSR sequences. It would be correct to rewrite that SNPs and SSRs are useful markers in GWAS.

Response: Thank you for your correction. We revised as you suggested.

2, 112. Paragraph structure is a major problem.  This is easiest to address using a key sentence to introduce the content of each paragraph. In scientific writing, the Key sentence is usually the first but can be the second sentence in the paragraph. The rest of the paragraph should support the statement in the key sentence.

Response: Thank you for your constructive comments and informative information we considered in the revised version.

Line 112, if moved to the first sentence Line 112 would make a good Key sentence for this paragraph. Line 84, is not. The paragraph is NOT devoted to evidence that “(SNPs) are more accurate and powerful in numerous perspectives of research. “

Response: Revised as you suggested.

Line 117. Delete Nevertheless

Response: Deleted.

  1. They? The statistics are not adequately described.  The comparisons in 1 C, 1D, and 1E are not clear.  All have the same captions, Control and SBV. It would really help if the X-axis label included the comparisons in each. Control and SBV labels alone are not helpful at all.  

Response: Thank you for your comments. “They?” was a typo revised.

For the figures of survival curves, we added the title, also the legends and caption were revised according to your comments.

If I read figure 1 correctly, there are an undescribed number of colonies that were assayed for presence or absence of 7 common honeybee viruses (primer sequences not given).  Number pairs above each bar may relate to the number of colonies assayed, but the numbers are hard to read,  not explained, and not helpful. Figure 1A shows that colonies were detected with 1) no virus, 2) SBV virus alone, 3) DWV virus alone, 4) both SBV and 5) DWV, SBV, DWV and IPV.  Figure I(B) compares infected and uninfected (Healthy vs SBV alone). Other virus effects not related to the paper are not shown.  The reason for the assay is to identify healthy colonies and colonies infected only by SBV. Because these are the only colonies studied further, it is important the total number of colonies in the two classes is given in the figure and text.

Response: Thank you for your comments. Actually, we screened out 80 colonies from different geographic locations against 7 common bee viruses, and that work was published before. Finding healthy colonies was a challenge in the present study.

In the current study, 8 colonies were screened in Minhou county; we found three healthy colonies out of 8 were healthy and without detectable diseases and used for larval rearing and genome re-sequencing in the first SNP validation as well. In the method of colony selection, to prevent the repetition of all the processes of virus detection, we mentioned and cited the paper. All 7 common bee virus primer sequences are available on that paper. The other primers we used are already attached in the list of primers table in the supplementary files. The legends and description results of screening colonies were revised and clearly stated.

200-202. Were the re-sequenced colonies those identified in Figure 1(A) as healthy or infected by SBV alone.  What number of each were sequenced? Delete “yielding” Delete or explain what is meant by “clean” data. “Clean is described in methods. Delete from here. A word that is often used to mean “clean” is vetted.  The phrase  “an average ratio of the samples to the reference genome of 87.92%”  makes no sense to me.  What is meant by ratio here?

Response: Thank you for your comments. Yes, 90 samples re-sequenced were from three colonies in Figure 1A. The three colonies from Figure 1(A) selected for larval rearing and re-sequencing were healthy and without any detectable diseases symptom. Each colony was composed of 30 samples (15 resistant and 15 susceptible larvae) described in the materials and methods section in the colony selection section.

Deleted. “Clean” data here means we got clean data and used in the mapping after filtering raw data. We took an average from all sample mapping percentages. It was revised as you suggested.

208 Typo? is rather than as?

Response: Revised.

209 Delete “according to statistical analysis” The analysis is undescribed as such and probably not a statistical analysis, but rather a numerical analysis. If so, that need not be stated.  Re: Table 1 references. If this comes from the “Statisitcal analysis” mentioned in line 209, that analysis must be described.

Response: Thank you for your comments. The statistical analysis was deleted and revised.

Note: Actually, the data in Table 1 we provided is giving a general overview of the summarized results from different statistical analyses or numerical data results from re-sequences data quality, filters, mapping up to identified SNP, indels, annotation, signature regions selection and identification of the SNP and genes to facilitate for readers. Cause of too big data, we cannot be placed it in the main body; even in the supplementary, we prefer to address them in a total. Please look at the tables in the appendix and additional files for details. May you also look at the value of significance predicted genes we uploaded the excel tables in the supplementary files. Please refer to other tables in the main body or supplementary files.

To be more specific, table 1 and the last column were deleted because we have/had that results in the supplementary file; also, table 2 from other statistical analyses was deleted, and only the selected region was kept and revised.

218 Delete either “we identified” or “were detected”. This can read “that were detected” but that addition requires a description of method of detection. The pronoun “It” refers to what?  Write out in full.

Response:  Deleted.

221-222 hererozygotes homozygotes

  1. Hetero ratio should be rewritten as heterozygosity.

Response:  Revised.

226 “Around 2.68% of heterozygosity, SNP” makes no sense.  The comma should be deleted and the phrase rewritten to make the meaning clear.  I read this to mean, The proportion of heterozygous SNPs in the R was 2.68 compared to XXX in the S.

Response: Thank you for your concern. Actually, 2.68% of heterozygosity was calculated from the whole samples’ average percentage of heterozygosity between R and S, e.g., 59.56%-56.88% = 2.68%. We kept the first calculation, and the second one was deleted.

  1. Bad English. I read this to mean, The other two possible mutant types did not differ in frequency.

Response: Revised.

  1. Delete “Form”

Response: Deleted.

252 This mutation must be explained, or the sentence deleted.

Response: Revised.

278 nonsynonymous mutants is a reference to codons, not genes.

Response: Thank you for your information.

280 Typo?  Between the reference “and” the sample.

Response: Revised.

297-8 fragment. No verb. Was based?  Are shown?

Response: Revised.

308-309. Fragment. No verb and I can’t guess the verb.

Response: Revised.

  1. Delete “before we used”. That word order works in German, but not in English.

Response: Deleted.

667-668 a fragment and  a strange reference to “just now”.  If the period is removed from the fragment and “just now is replaced with “from which”, this reads OK, assuming the revised sentence is true.

Response: Thank you for your correction. We revised as you suggested.  

  1. A remarkable observation.  It suggest mutation in transcription factor, rather than an altered protein. A bit more comment on this could be ground breaking.

Response: Thank you for your information.

697-698. Needs to be rewritten as two clear statements.  This appears to say that larvae thought to be susceptible, when  the they mature are found to be  resistant. I do not recall reading that in results and do not know what this contributes to the discussion.

Response: Thank you for your suggestion. There was a verb missing revised, and the two statements were separated. I would like to give you more explanation here for the second part of your question. As you know, SBV can infect honeybees at different stages, e.g., (eggs, larva, prepupa, and adults bee); however, SBV are not able to infect pupae, or in another statement, we can say when the larvae are susceptible to SBV before to convert to pupae stages that might be infected at early stages (larva) or late stages (prepupae) and might die before pupae stages. In order to correctly select resistant samples, the pupae were sampled as resistant to SBV.

  1. Typo? Close? 

Response: Revised.

  1. It would be good to include this in the introduction.  Lacking this, the reader must guess why A. cerana was selected for the study.

Response: Thank you for your suggestion. That sentence moved to introduction.

  1. Themselves?  Is this a reference to Workers?

Response: Thank you for your correction ‘worker bees’ revised.

826 Delete “as needed” which implies not used.  Write ”as in”  this study.

Response: Revised as in.

  1. “of 90 accessions” is a dangling phrase. It is not clear what the phrase references.

Response:  All th accession phrases were replaced with sample/samples.

  1. Sentence fragment.

Response: Thank you revised.

967 delete period after dehydration.

Response: Deleted.

  1. Statistical analysis of survival functions. The paper will be much easier to understand if  M& M follows Introduction and Conclusions follow discussion. The problem is that as  written, the Results too often repeat what was done, while the question of why it was done (that should appear in Results) is imbedded  in methods. The Reader trying to make it through Results has a very incomplete and confusing view of what was done, and no idea of why it was done.

Finally, I strongly suggest that “based on selective sweep analysis” be deleted be deleted from the title. The re-sequencing is not based on selective sweep analysis and the discovery of markers and candidate genes is not based on more than selective sweep analysis.

Response: Statistical analysis of survival functions was deeply revised and edited. We tried to consider your suggestions and followed up on your instructions.

Once again, thank you for your correction, constructive comments, and valuable suggestions. The authors would like to thank you for taking the time and effort to carefully review, figure out errors, and provide clear comments and suggestions. We tried our best to give you a response to each question and revise/delete all the questions you have mentioned and correct our mistakes one by one. We were really happy with your review and learned a lot from you.

                                                                             Thanks!

Reviewer 3 Report

This is an extremely well done set of experiments on a very important topic - identification of genomic polymorphisms associated with resistance to viral infection. I'm unsure of the availability of effective interventions for viral infection in colonies in China, but I am keenly aware that none are available for beekeepers here in the US and this study will stand out as an example of how other researchers should approach identification of resistance alleles. I also want to commend the authors on how thorough they were in reporting their data in the form of supplementary material - it can often be difficult to get authors to supply even a single table. I do have a few minor comments:

1. To be frank, the paper was very difficult to read at times which contributed to my tardiness on this review. This is not meant to reflect on the authors at all (since, I cannot read or write in Chinese, after all) but the paper requires significant editing of the English language to improve clarity.
2. In some cases, labels are highly ambiguous. For example on Table 1, the second to last column has what I believe are statistics labeled "Fst_R_vs0.99" etc. It took me a very long time to understand what these labels meant and I am still not certain I understand. Can the authors please try to workout a clearer labeling system? This is also a problem on other tables and figures. 
3. It's not clear how colonies were designated as "resistant" or "susceptible". Was it know a priori which colonies were a resistant or susceptible stock to SBV? Or were colonies labeled resistant or susceptible post hoc following the mortality curve analyses?
4. Speaking of, in Figure 1 - did the authors compare the survival curves of the resistant (1D) and susceptible (1E) to one another? Are those survival differences significant at the 0.0001 level?
5. Very minor - but the concluding paragraph of the introduction could be expanded or rewritten to summarize the findings.

I am looking forward to reading this paper in print.

Author Response

Open Review

Comments and Suggestions for Authors

This is an extremely well done set of experiments on a very important topic - identification of genomic polymorphisms associated with resistance to viral infection. I’m unsure of the availability of effective interventions for viral infection in colonies in China, but I am keenly aware that none are available for beekeepers here in the US and this study will stand out as an example of how other researchers should approach identification of resistance alleles. I also want to commend the authors on how thorough they were in reporting their data in the form of supplementary material - it can often be difficult to get authors to supply even a single table. I do have a few minor comments:

Response: The authors really appreciated your praise and interest in our work. According to literature, scientists put a lot of effort into managing viral infections in honeybees, especially in the sacbrood virus (SBV) around the globe. However, SBV is still a severe disease in honeybee colonies in China and abroad. No treatment, effective and easy method was available until now. Therefore, we conducted this study to solve this problem. We will not discuss here and show what was done before or what we have done. Based on previous studies, our research group developed another SNP molecular marker related to chalkbrood disease resistance in Apis melliferra and bred resistant queen, namely Queen NongDa-1, and identified SNP related to high royal jelly production. We hope other researchers use this approach to identify resistance alleles in another threatened factor in Apis millefleur, such as the deformed wing virus. That would be a valuable and interesting topic in perspective research.

We appreciate your concern.We have attached all the important data to the supplementary files, which will be made accessible after the download link is created, and published; however, in each part of the data you need something, please do not hesitate to contact the appropriate author, if there is any request, we are happy to provide you the materials requested and feel free to answer your questions.

To be frank, the paper was very difficult to read at times which contributed to my tardiness on this review. This is not meant to reflect on the authors at all (since, I cannot read or write in Chinese, after all) but the paper requires significant editing of the English language to improve clarity.

Response: Thank you for your suggestion; we have already applied paid English editing services we hope to be improved in the revised version.

1, In some cases, labels are highly ambiguous. For example on Table 1, the second to last column has what I believe are statistics labeled “Fst_R_vs0.99” etc. It took me a very long time to understand what these labels meant and I am still not certain I understand. Can the authors please try to workout a clearer labeling system? This is also a problem on other tables and figures. In our results, the significance of regions was first detected with the two approaches, Fst and Pi; even with other methods independently, each comparison has its own results to consider the narrow range we consider the combined Fst and Pi, also based on this narrow range it easy to select the candidate SNP and gene for validation table 1 revised and some figures we plotted from analyzed data were removed we considered the last selected regions.

Response: In fact, the data in Table 1 we provided is giving a general overview of the results from re-sequences data quality, mapping, filters, up to identified SNP, indels, annotation, signature regions selection, and identification of the SNPs and genes based in summed form. Cause of too big data, we could not attach them in the supplementary files; we just put the total number and plotted the analyzed data tables in the form of figures. May you have a look at Figure 3, how the intersection and combined Fst and Pi significance regions were detected. May you also look at the value of significance predicted genes we uploaded the excel tables in the supplementary files, which is clearly described the scaffold, regions, and Fst values.

2, It’s not clear how colonies were designated as “resistant” or “susceptible”. Was it know a priori which colonies were a resistant or susceptible stock to SBV? Or were colonies labeled resistant or susceptible post hoc following the mortality curve analyses?

Response: Thanks for your question, as described in the materials and methods. The susceptible colonies were directly selected based on the visible symptoms of SBV in the field colonies. The colonies were bought from local beekeepers and established in the bee apiary experimental place to conduct larval rearing in vitro based on the mortalities rate and genotyped susceptible and resistant larvae using current identified SNP with detection of susceptibility alleles (C and T) the susceptibility of colonies was determined. Following the resistance, colonies were detected during larval rearing and genotyped samples during SNP validation in the field with high G and C allele frequency in the two SNPs, as discussed in the discussion. It is very easy to distinguish susceptibility colonies on the field based on SBV symptoms and during larval rearing in vitro after inoculation with the virus.

3, Speaking of, in Figure 1 - did the authors compare the survival curves of the resistant (1D) and susceptible (1E) to one another? Are those survival differences significant at the 0.0001 level?

Response: Thanks, no, in Figures 1D and 1E are just the merged tree replicates of three colonies. Comparing the survival of control (uninoculated) with the survival curve of treatment groups (inoculated with SBV), three susceptible and three resistant colonies were compared with three replicates, as shown in two curves and two figures. The survival curves of larvae raised in vitro between treatment and control were < 0.0001, with a p-value < 0.01 defined as significantly different in all comparisons. If we look at the two figures, we can clearly see the difference in the survival of resistant and susceptible colonies compared to the treatment and control. The purpose of that comparison was to see the symptom of SBV in the treated groups compared with no inoculated group (control) because sometimes larvae died; we don’t know the virus or environmental factors caused that, and sometimes the virus maybe inactivated and lost their infectious/contiguous. Thus, this comparison was enough in our experiment to distinguish the infected and uninfected larvae better. To consider the reader and clarify, we revised the caption, added titles to the figures, and revised the description in the legend. 

4, Very minor - but the concluding paragraph of the introduction could be expanded or rewritten to summarize the findings.

I am looking forward to reading this paper in print.

Response: Thanks for your suggestion. The concluding paragraph of the introduction was a little bit expanded.
